# Phenylhydrazone-based endoplasmic reticulum proteostasis regulator compounds with enhanced biological activity

**Gabriel M Kline[1], Lisa Boinon[2], Adrian Guerrero[1], Sergei Kutseikin[3], Gabrielle Cruz[3], Marnie P Williams[2], Ryan J Paxman[1], William E Balch[3], Jeffery W Kelly[1]\*, Tingwei Mu[2]\*, R Luke Wiseman[3]\***

[1]Department of Chemistry, The Scripps Research Institute, San Diego, United States; [2]The Department of Physiology and Biophysics, Case Western Reserve University School of Medicine, Cleveland, United States; [3]Department of Molecular and Cellular Biology, The Scripps Research Institute, San Diego, United States

## eLife Assessment

This study reports the **important** development and characterization of next-generation analogs of the molecule AA263, which was previously identified for its ability to promote adaptive ER proteostasis remodeling. The evidence supporting the conclusions is **convincing**, with rigorous assays used to benchmark the changes in potency and efficacy of the AA263 analogs as well as AA263 targets. The ability of AA263 analogs to restore the loss of function associated with disease-associated proteins prone to misfolding will be of interest to pharmacologists, chemical biologists, and cell biologists, as well as those working on protein misfolding disorders.

**\*For correspondence:**
jkelly@scripps.edu (JWK);
txm210@case.edu (TM);
wiseman@scripps.edu (RLW)

**Abstract** Pharmacological enhancement of endoplasmic reticulum (ER) proteostasis is an attractive strategy to mitigate pathology linked to etiologically diverse protein misfolding diseases. However, despite this promise, few compounds have been identified that enhance ER proteostasis through defined mechanisms of action. We previously identified the phenylhydrazone-based compound AA263 as a molecule that promotes adaptive ER proteostasis remodeling through mechanisms including preferential activation of the ATF6 signaling arm of the unfolded protein response (Plate et al., 2016). However, the protein target(s) of AA263 and the potential for further development of this class of ER proteostasis regulators had not been previously explored. Here, we employ chemical proteomics to demonstrate that AA263 covalently targets a subset of ER protein disulfide isomerases, revealing a potential molecular mechanism for the activation of ATF6 afforded by this compound. We then use medicinal chemistry to establish next-generation AA263 analogs showing improved potency and efficacy for ATF6 activation, as compared to the parent compound. Finally, we show that treatment with these AA263 analogs enhances secretory pathway proteostasis to correct the pathologic protein misfolding and trafficking of both a destabilized, disease-associated α1-antitrypsin (A1AT) variant and an epilepsy-associated GABA$_A$ receptor variant. These results establish AA263 analogs with enhanced potential for correcting imbalanced ER proteostasis associated with etiologically diverse protein misfolding disorders.

## Introduction

Nearly one-third of the human proteome is targeted to the endoplasmic reticulum (ER) for folding and trafficking to downstream secretory environments including the plasma membrane, lysosome, and the extracellular space (*Hipp et al., 2019*; *Jayaraj et al., 2020*; *Araki and Nagata, 2011*). The folding and trafficking versus degradation of these secretory proteins is decided by a process termed ER quality control, wherein ER-localized chaperones and folding factors engage proteins within the ER to facilitate their folding into a folded, trafficking-competent conformation (*Hipp et al., 2019*; *Jayaraj et al., 2020*; *Kaushik and Cuervo, 2015*). Proteins unable to attain a folded conformation in the ER through interactions with ER chaperones are instead recognized by ER-localized degradation factors that promote their targeting to degradation pathways such as ER-associated degradation by the proteasome or ER-phagy accomplished by the lysosome (*Kaushik and Cuervo, 2015*). Through this ER quality control process, cells promote the trafficking of folded, functional proteins through the secretory pathway and prevent the accumulation of non-native or aggregation-prone conformations within the ER or in downstream secretory environments (*Labbadia and Morimoto, 2015*; *Wiseman et al., 2022*).

Despite the efficacy of ER quality control, failure of this process is implicated in the onset and pathogenesis of numerous diseases, collectively referred to as protein misfolding diseases (*Mesgarzadeh et al., 2022*). These diseases are primarily associated with destabilizing mutations within a secretory protein that hinder ER quality control machinery, ultimately leading to toxic protein aggregation and/or loss-of-function pathologies. For example, the aberrant secretion and toxic extracellular aggregation of destabilized, aggregation-prone variants of amyloidogenic proteins including transthyretin (TTR) or immunoglobulin light chain (LC) are implicated in the onset and pathogenesis of protein aggregation diseases such as the TTR amyloidoses and light chain amyloidosis (*Desport et al., 2012*; *Sekijima, 2014*). Alternatively, mutations in subunits of the gamma-aminobutyric acid type A (GABA$_A$) receptors prevent the proper folding, assembly, and trafficking of these proteins to the plasma membrane. This loss-of-function phenotype has been implicated in numerous neurological disorders, including genetic epilepsy and neurodevelopmental delay (*Braat and Kooy, 2015*; *Fu et al., 2022*). Other proteinopathies, such as alpha-1-antitrypsin (AAT) deficiency, can involve both toxic protein aggregation and loss-of-function pathologies. In this disease, destabilized AAT variants (e.g., PIZZ) can lead to both gain of toxic function pathology through intra-ER retention causing liver pathology and loss-of-function pathology associated with poorly functional secreted AAT leading to chronic obstructive pulmonary disease in the lung (*Lomas and Mahadeva, 2002*; *Silverman and Sandhaus, 2009*).

The central involvement of ER quality control in the onset and pathogenesis of protein misfolding diseases has suggested that developing strategies to enhance ER quality control offers a potential opportunity to broadly mitigate pathologies linked to the aberrant folding or degradation of proteins targeted to the ER (*Plate and Wiseman, 2017*). One attractive strategy to regulate ER quality control is through targeting the unfolded protein response (UPR) – the predominant stress responsive signaling pathway that regulates ER quality control in response to pathologic ER insults (*Ron and Walter, 2007*; *Walter and Ron, 2011*; *Preissler and Ron, 2019*). The UPR comprises three integrated signaling pathways activated downstream of the ER membrane proteins IRE1, PERK, and ATF6 (*Hetz et al., 2020*). ER stress activates the UPR leading to downstream transcriptional and translational signaling that functions to both alleviate the ER stress and enhance ER function. ER quality control is primarily regulated by the UPR-associated transcription factors XBP1s (downstream of IRE1) and ATF6 (a cleaved product of full-length ATF6) (*Adachi et al., 2008*; *Yoshida et al., 2001*). These transcription factors induce expression of overlapping, but distinct, sets of ER chaperones, folding enzymes, and degradation factors that remodel the composition and the capacity of ER quality control pathways by creating emergent functions (*Adamson et al., 2016*; *Shoulders et al., 2013*). Consistent with this, stress-independent activation of XBP1s, or especially, ATF6 has been shown to correct pathologic imbalances in ER quality control implicated in the pathogenesis of many different protein misfolding diseases, including those indicated above (*Chen et al., 2014*; *Cooley et al., 2014*; *Plate et al., 2019*).

The potential to improve ER quality control in protein misfolding diseases through activation of ATF6 led to a significant interest in developing pharmacologic approaches that selectively induce activation of this UPR transcriptional program. Toward that aim, we previously used high-throughput screening to identify first-in-class ER proteostasis regulator compounds, such as compound AA147,

that preferentially activate the ATF6 signaling arm of the UPR (*Plate et al., 2016*). We showed that AA147 activates ATF6 through a mechanism involving metabolic conversion of the prodrug AA147 to an electrophilic quinone methide on the ER membrane that covalently modifies a subset of ER protein disulfide isomerases (PDIs) (*Paxman et al., 2018*). This leads to an increase in reduced ATF6 that can traffic to the Golgi for proteolytic release of the active ATF6 transcription factor domain (*Higa et al., 2014*; *Koba et al., 2020*; *Oka et al., 2019*; *Oka et al., 2022*). Intriguingly, treatment with AA147 improves ER quality control for many different destabilized, disease-associated proteins linked to protein aggregation and loss-of-function protein misfolding diseases including AAT deficiency and GABA$_A$ receptor-associated epilepsy, as well as protein aggregation-associated degenerative diseases such as TTR amyloidosis and immunoglobulin light chain amyloidosis (*Plate et al., 2016*; *Blackwood et al., 2019*; *Sun et al., 2023*; *Wang et al., 2022a*). As predicted based on the mechanism of action of AA147, these benefits can be attributed to ER proteostasis remodeling induced by either direct PDI modification and/or by ATF6 activation, demonstrating the broad potential for this approach to influence ER proteostasis of multiple different disease-relevant proteins (*Rius et al., 2021*; *Rosarda et al., 2024*). However, despite this promise, the continued development of AA147 has proven challenging, as AA147 has been shown to be largely recalcitrant toward scaffold modifications aimed at improving its potency, selectivity, and efficacy for ATF6 activation (*Paxman et al., 2018*; *Kline et al., 2024*).

In addition to finding AA147, our original high-throughput screen also identified the phenylhydrazone compound AA263 as a compound that preferentially activates the ATF6 arm of the UPR (*Plate et al., 2016*). However, the mechanism of AA263-dependent ATF6 activation and the potential for the development of improved analogs based on this scaffold by medicinal chemistry have not been previously explored. Herein, we show that AA263, like AA147, covalently targets ER PDIs, providing a potential mechanism to explain the preferential ATF6 activation afforded by this compound. Intriguingly, we also show that modification of the AA263 B-ring at the *para* position allows for the establishment of compounds with improved selectivity and efficacy for ATF6 activation. Taking advantage of this, we establish next-generation AA263 analogs that show improved potency and efficacy in comparison to the parent AA263. Further, we demonstrate that these enhanced AA263 analogs correct ER quality control defects in cellular models expressing disease-relevant proteins including the Z variant of AAT and epilepsy-associated variants of GABA$_A$ receptors. Collectively, these results establish AA263 and its associated analogs as ER proteostasis regulators with enhanced potential for further development, expanding the toolbox of compounds suitable for targeting ER quality control in the context of diverse protein misfolding diseases.

## Results

### AA263 covalently modifies a subset of ER PDIs

ER proteostasis regulators including AA147 and AA132 activate the ATF6 arm of the UPR through a mechanism involving their metabolic activation to a quinone methide and subsequent covalent targeting of ER PDIs (*Paxman et al., 2018*; *Kline et al., 2023*). Intriguingly, 2-hydroxyphenylhydrazones are known to tautomerize to a quinone methide (AA263-QM), suggesting that AA263 could similarly activate ATF6 through spontaneous quinone methide electrophile generation and covalent PDI targeting (*Figure 1A*), without the requirement of metabolic activation (*Ifa et al., 2000*). Interestingly, co-treatment with β-mercaptoethanol (BME), which acts as both an exogenous nucleophile and modulator of cellular redox potential, or the antioxidant resveratrol decreased AA263-dependent activation of the ERSE-FLuc reporter (*Figure 1B, C*) suggesting that an ER enzyme oxidative activation of AA263 to an unknown electrophile could also be playing a role in PDI conjugation. Consistent with this, synthesis of an AA263 analog lacking the hydroxyl moiety on the A-ring (AA263-1; *Figure 1— figure supplement 1A*) does not activate the ATF6-selective ERSE-FLuc reporter in HEK293 cells. These results support a model whereby AA263 activates ATF6 through a mechanism involving both enzymatic oxidation and direct tautomerization to an AA263-QM and subsequent covalent modification of predominantly ER protein targets (*Figure 1A*).

To monitor covalent protein labeling by AA263, we generated an AA263 analog with replacement of the nitro group on the B-ring with an alkyne amenable for affinity enrichment experiments (AA263[yne], *Figure 1D*). We found that this analog robustly activated the ERSE-FLuc reporter, demonstrating an ~2-fold increase in potency, as compared to AA263 (*Figure 1E*). Co-treatment with the

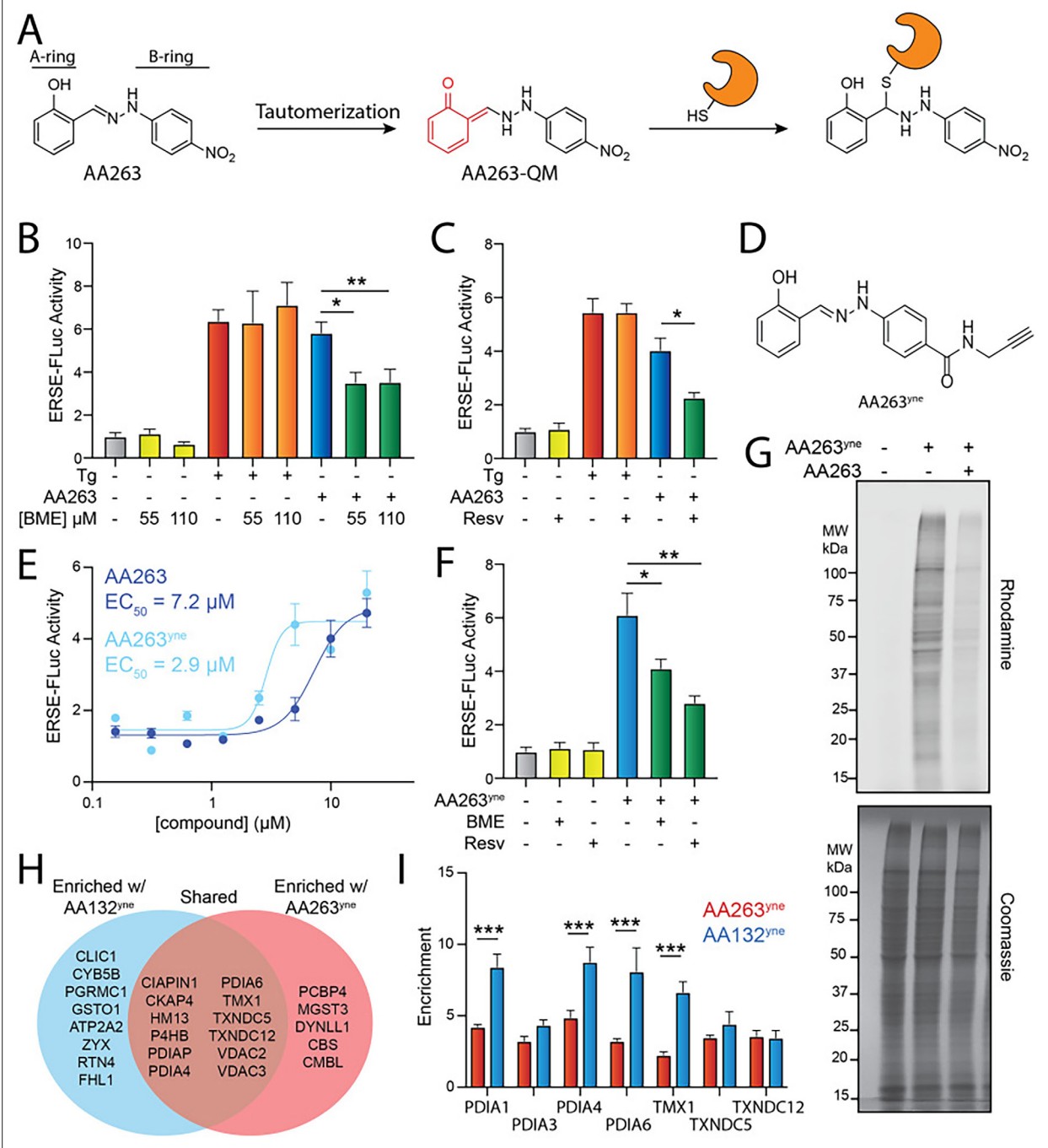

**Figure 1.** AA263 covalently modifies protein disulfide isomerase (PDI) family members. (**A**) Mechanism of AA263 metabolic activation and covalent protein modification. (**B**) Activation of the ERSE.FLuc ATF6 reporter in HEK293T cells treated for 18 hr with AA263 (10 µM) or thapsigargin (Tg, 500 nM) in the presence or absence of β-mercaptoethanol (BME; 55 or 110 µM). Error bars show SEM for $N > 6$ replicates. *$p < 0.05$, **$p < 0.01$ for unpaired $t$-test. (**C**) Activation of the ERSE.FLuc ATF6 reporter in HEK293T cells treated for 18 hr with AA263 (10 µM) or Tg (500 nM) in the presence or absence of resveratrol (2.5 µM). Error bars show SEM for $N > 6$ replicates. *$p < 0.05$ for unpaired $t$-test. E. (**D**) Structure of AA263$^{yne}$. (**E**) Activation of the ERSE. FLuc ATF6 reporter in HEK293T cells treated for 18 hr with the indicated dose of AA263 or AA263$^{yne}$. Error bars show SEM for $n = 3$ replicates. The $EC_{50}$ is shown. (**F**) Activation of the ERSE.FLuc ATF6 reporter in HEK293T cells treated with AA263$^{yne}$ (10 µM) in the presence or absence of BME (55 µM) or resveratrol (2.5 µM). *$p < 0.05$, **$p < 0.01$ for unpaired $t$-test. (**G**) Representative SDS–PAGE gel of Cy5-conjugated proteins from HEK293T cells treated for 4 hr with vehicle (0.1% DMSO), AA263$^{yne}$ (5 µM), or the combination of AA263$^{yne}$ (5 µM) and AA263 (20 µM). Coomassie-stained gel is shown below. (**H**) Venn diagram of identified targets of AA132$^{yne}$ and AA263$^{yne}$. Hits defined as proteins with a significant fold change greater than 3 ($p < 0.01$) that were identified in two independent biological experiments. (**I**) TMT reporter ion enrichment ratio of select PDIs from comparative chemoproteomic

*Figure 1 continued*

experiment in HEK293T cells treated with the indicated compound relative to DMSO ($n$ = 8 biological replicates). ***$p < 0.005$ for a two-way ANOVA. See also *Figure 1—source data 1*.

The online version of this article includes the following source data and figure supplement(s) for figure 1:

**Source data 1.** Excel spreadsheet showing raw data used to generate panels *Figure 1B, C, E, F, and I*.

**Source data 2.** Uncropped gel shown in *Figure 1G*.

**Source data 3.** Uncropped gel of *Figure 1G* with the same annotations shown in *Figure 1G*.

**Figure supplement 1.** AA263 covalently modifies protein disulfide isomerase family members.

**Figure supplement 1—source data 1.** Excel spreadsheet showing raw data used to generate *Figure 1—figure supplement 1A and B*.

**Figure supplement 1—source data 2.** Uncropped gels shown in *Figure 1—figure supplement 1C and D*.

**Figure supplement 1—source data 3.** Uncropped gels shown in *Figure 1—figure supplement 1C and D* with the same annotations shown in *Figure 1—figure supplement 1C and D*.

selective ATF6 inhibitor Ceapin-A7 (CP7) (*Torres et al., 2019*; *Gallagher and Walter, 2016*; *Gallagher et al., 2016*) blocked AA263$^{yne}$-dependent increases in ERSE-Fluc activity, confirming that this effect can be attributed to ATF6 activation (*Figure 1—figure supplement 1B*). We also confirmed that the AA263$^{yne}$-dependent activation of the ERSE-FLuc reporter was inhibited by co-treatment with BME or resveratrol (*Figure 1F*). We treated HEK293 cells with AA263$^{yne}$ and visualized protein modification by appending a Cy5-azide fluorophore to the terminal alkyne of conjugated proteins using Cu(I)-catalyzed azide-alkyne cycloaddition (CuAAC). AA263$^{yne}$ showed robust protein labeling in HEK293 cells upon in situ compound treatment (*Figure 1G*). Co-treatment with excess AA263 reduced labeling by AA263$^{yne}$, confirming that AA263$^{yne}$ and AA263 label the same subset of the human proteome. Further, co-treatment with BME or resveratrol also reduced AA263$^{yne}$ proteome labeling (*Figure 1—figure supplement 1C*). Intriguingly, unlike AA147$^{yne}$, treatment of lysates with AA263$^{yne}$ showed mild proteome labeling, indicating that some AA263-dependent protein labeling may proceed through direct tautomerization to the electrophilic QM species (*Figure 1—figure supplement 1D*).

We next sought to identify proteins modified by AA263 using Tandem Mass Tag (TMT)-based quantitative proteomics. We previously showed that preferential ATF6 activation afforded by compounds such as AA147 and AA132 is attributed to the extent of compound-dependent modification of PDIs (*Kline et al., 2023*). Thus, we compared the extent of proteome labeling observed with AA263$^{yne}$ labeling to that observed with AA132$^{yne}$ – a compound that globally activates the UPR through covalently modifying a large number of PDIs (*Kline et al., 2023*). For these comparative chemoproteomic experiments, we treated HEK293 cells with vehicle, AA263$^{yne}$, or AA132$^{yne}$ and then used CuAAC to append biotin-azide to modified proteins. Proteins were then enriched with streptavidin, digested with trypsin digestion, labeled with TMTs, and analyzed by LC–MS/MS (*Chan et al., 2021*; *Zhang and Elias, 2017*). Modified proteins were defined by a threefold enrichment in compound-treated cells ($p < 0.01$), as compared to vehicle-treated cells. We identified 17 proteins covalently modified by AA263$^{yne}$, as compared to 20 proteins modified by AA132$^{yne}$ (*Figure 1H*). Intriguingly, 12 proteins were shared between these two conditions, including 7 different ER-localized PDIs (*Figure 1H*). This includes PDIs previously shown to regulate ATF6 activation including TXNDC12/ERP18 (*Oka et al., 2019*; *Oka et al., 2022*). These results are similar to those observed when comparing proteins modified by the preferential ATF6 activating compound AA147$^{yne}$ and AA132$^{yne}$ (*Kline et al., 2023*). Further, we found that the extent of labeling for PDIs including PDIA1, PDIA4, PDIA6, and TMX1, but not TXNDC12, showed greater modification by AA132$^{yne}$, as compared to AA263$^{yne}$ (*Figure 1I*). Similar results were observed for AA147$^{yne}$ (*Kline et al., 2023*). This suggests that, like AA147, the preferential activation of ATF6 afforded by AA263 is likely attributed to the modifications of a subset of multiple different ER-localized PDIs by this compound. Four of the 5 proteins that formed conjugates with AA263$^{yne}$, but not AA132$^{yne}$, are cytosolic proteins, suggesting that tautomerization to an AA263-QM may be occurring as a minor quinone methide formation pathway. However, spontaneous tautomerization to the quinone methide of AA263 seems unlikely to be the main pathway of electrophile formation from AA263 given the remarkable preference of AA263 to react with ER-localized proteins, which is not a general feature of other small molecule electrophiles (*Backus et al., 2016*; *Paxman et al., 2018*). This specificity for ER proteins instead suggests the localized generation of AA263 quinone methides at the ER membrane, likely through metabolic activation by different ER-localized oxidases, which has

previously been shown to contribute to the selective modification of ER proteins afforded by other compounds such as AA147 (**Kline et al., 2023**).

## Structure–activity relationship studies identify AA263 analogs with improved potency and efficacy

The increased potency observed for AA263$^{yne}$, as compared to AA263, suggests that the B-ring of AA263 may be amenable to medicinal chemical modification for modulating the activation of ATF6 by this class of compounds (**Figure 1E**). Further, nitro groups are known to alter mitochondrial function, so removal of this functional group could improve cellular tolerability and reduce cytotoxicity associated with two successive 1-electron reductions of the nitro group to afford cytotoxic aromatic nitroso compounds (**Coe et al., 2007**; **Nepali et al., 2019**). Toward that aim, we synthesized a panel of AA263 derivatives comprising a heteroaromatic B-ring and aryl B-rings with varying functional groups largely but not exclusively appended at the *para* position of the B-ring (**Figure 2A**). We screened these compounds for ATF6 activation using HEK293 cells stably expressing the ATF6-selective ERSE-Fluc reporter (**Figure 2B**). This effort demonstrated that ATF6 activators are tolerant of B-ring *para* amide and ester functional groups, reflected by AA263$^{yne}$ and AA263-5 being the two most active AA263 analogs. Interestingly, the *para* carboxylate substitution affords an inactive compound (AA263-4). We confirmed that AA263$^{yne}$ and AA263-5 induced expression of the ATF6 target gene *HSPA5/BiP* in HEK293 cells (**Figure 2—figure supplement 1A**) to levels higher than that observed for AA263, without increasing expression of the IRE1/XBP1s target gene *ERdj4* (**Figure 2—figure supplement 1A**) or the PERK/ISR target gene *CHOP/DDIT3* (**Figure 2—figure supplement 1A**). The inactive compound AA263-2 (**Figure 2A**) featuring a heterocyclic B-ring was used as a control and did not show activation of UPR target genes. ATF6 activators AA263$^{yne}$ and AA263-5 also both showed increased potency for ERSE-FLuc activation relative to AA263, further highlighting the improved activity of these two compounds (**Figure 2—figure supplement 1B**).

Next, we used RNAseq to further probe the selectivity for ATF6 activation in HEK293 cells treated with either vehicle, AA263, AA263$^{yne}$, or AA263-5. Initially, we monitored the activation of ATF6 and other arms of the UPR by comparing the expression of sets of genes regulated by these different UPR signaling pathways (**Grandjean et al., 2019**). Aligning with our qPCR results, AA263$^{yne}$ and AA263-5 increased expression of ATF6 target genes, in comparison to AA263 (**Figure 2C**, **Supplementary file 1**). These compounds also showed a modest increase in IRE1/XBP1s target genes, reflecting the known overlap between ATF6 and IRE1/XBP1 target genesets (**Shoulders et al., 2013**). In contrast, both compounds modestly reduced expression of PERK/ISR target genes. This may result from removal of the nitro group, as this geneset can be induced by mitochondrial dysfunction resulting from the presence of this moiety (**Lebeau et al., 2018**; **Verfaillie et al., 2012**). These results indicate that AA263$^{yne}$ and AA263-5 both increase adaptive ATF6 signaling, while maintaining or improving preferential selectivity for this specific arm of the UPR over the other two UPR arms. Importantly, we did not observe activation of gene sets regulated downstream of other stress-responsive signaling pathways such as the oxidative stress response or heat shock response in HEK293 cells treated with these other AA263 analogs (**Figure 2—figure supplement 1C**, **Supplementary file 1**). Further, Gene Ontology analysis showed that AA263$^{yne}$ and AA263-5 primarily induced expression of genes involved in biological pathways linked to ER proteostasis and UPR activation (**Supplementary file 2**; **Mi et al., 2013**). Collectively, these results indicate that AA263$^{yne}$ and AA263-5 both show enhanced activation of the ATF6 transcriptional program, without sacrificing the transcriptome-wide preferential selectivity for ATF6 activity observed for the parent compound AA263.

We next sought to evaluate whether we could further elaborate the *para*-amide substructure of the AA263$^{yne}$ scaffold by a medicinal chemistry strategy to provide additional enhancement of ATF6 activity. Toward that aim, we synthesized AA263$^{yne}$ analogs where the alkyne moiety was replaced with various aliphatic or aromatic substituent groups (**Figure 3A**). We then monitored activation of the ERSE-FLUC ATF6 reporter stably expressed in HEK293 cells treated with increasing doses of these analogs. This identified two analogs, AA263-15 and AA263-20, that showed higher levels of reporter activation, as compared to AA263 or AA263$^{yne}$, with increased potency (**Figure 3B, C**). Both AA263-15 and AA263-20 increased expression of the ATF6 target gene *BiP* at 6 hr after 5 µM treatment to levels greater than that observed for AA263$^{yne}$ in HEK293 cells (**Figure 3D**). However, these compounds did not significantly influence expression of the IRE1/XBP1s target gene *ERDJ4* or the PERK target gene

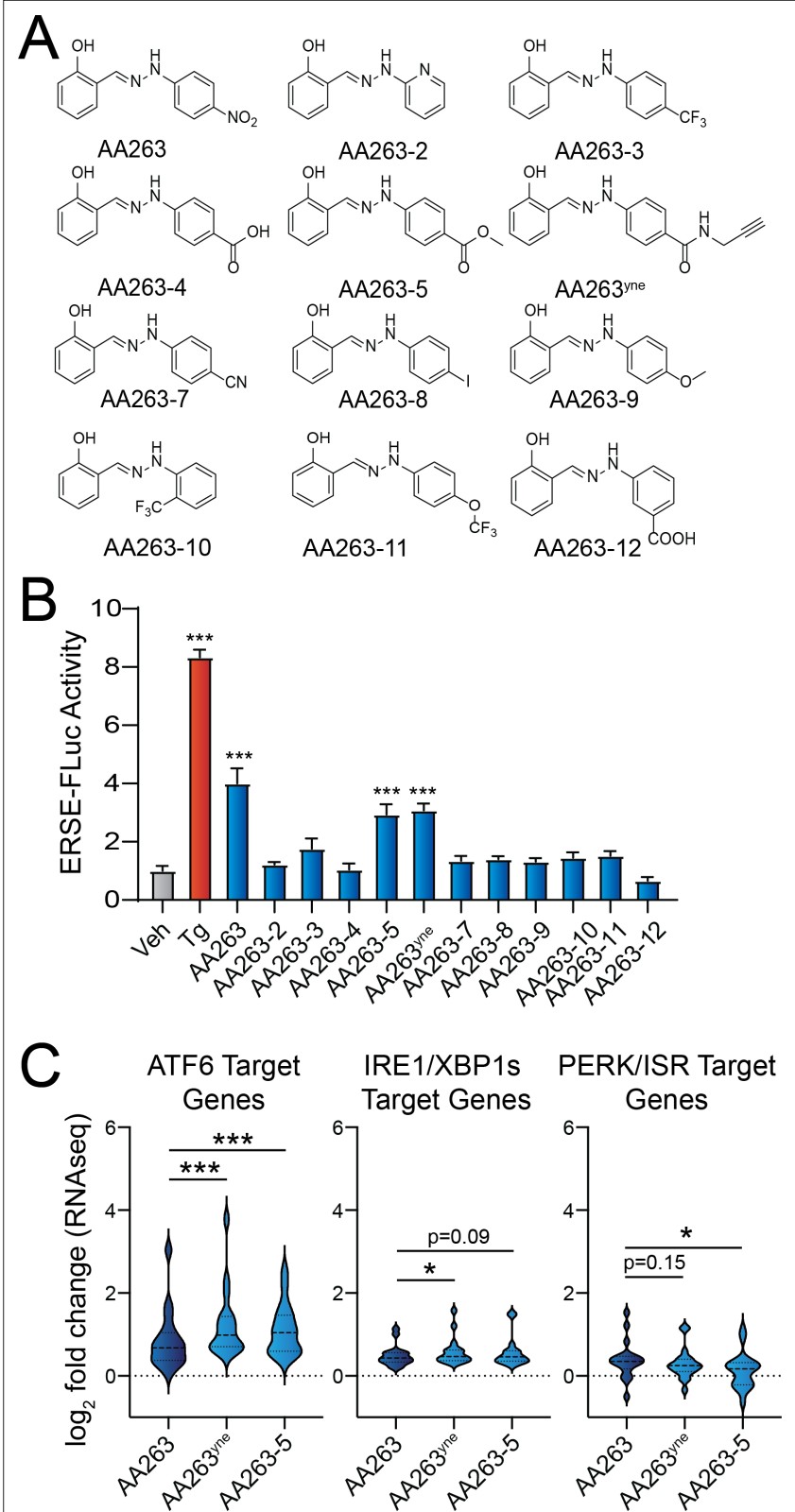

**Figure 2.** Identification of AA263 analogs that show enhanced ATF6 activation. (**A**) Structures of AA263 analogs. (**B**) Activation of the ERSE.Fluc ATF6 reporter in HEK293T cells reporter treated for 18 hr with vehicle, Tg (0.5 μM), or the indicated analog (10 μM). Error bars show SEM for $n$ = 3–6 biological replicates. ***p < 0.005 from one-way ANOVA. (**C**) Expression, measured by RNAseq, of gene sets comprising target genes regulated downstream of the

*Figure 2 continued on next page*

*Figure 2 continued*

ATF6 (left), IRE1/XBP1s (middle), or PERK/ISR (right) arms of the unfolded protein response (UPR) in HEK293T cells treated for 6 hr with 10 μM AA263, AA263$^{yne}$, or AA263-5. Full RNAseq data and genesets used in this analysis are shown in *Supplementary file 1*. *p < 0.05, ***p < 0.005 for one-way ANOVA. See also *Figure 2—source data 1*.

The online version of this article includes the following source data and figure supplement(s) for figure 2:

**Source data 1.** Excel spreadsheet showing raw data used to generate *Figure 2B*.

**Figure supplement 1.** Identification of AA263 analogs that show enhanced ATF6 activation.

**Figure supplement 1—source data 1.** Excel spreadsheet showing raw data used to generate *Figure 2—figure supplement 1A and B*.

---

*CHOP* in these cells (*Figure 3—figure supplement 1A*). Further, we confirmed that both AA263-15 and AA263-20 competed with proteome labeling afforded by AA263$^{yne}$ (*Figure 3—figure supplement 1B*), demonstrating that these compounds covalently targeted a similar subset of the proteome. These results demonstrate the potential for further developing AA263 analogs for preferential ATF6 activation and identify two compounds, AA263-15 and AA263-20, as compounds with improved activity relative to AA263 or AA263$^{yne}$.

## Enhanced AA263 analogs improve secretory proteostasis for the disease-associated Z variant of A1AT

We next sought to define the potential of AA263 analogs for promoting adaptive ER remodeling in cellular models of protein misfolding diseases. AAT deficiency is a complex, multi-tissue disorder caused by the pathologic aggregation of destabilized AAT variants in the ER of hepatocytes comprising the liver and the reduced ability of secreted mutant AAT to inhibit neutrophil elastase (NE), leading to lung dysfunction (*Lomas and Mahadeva, 2002*). Previously, we showed that treatment with ER proteostasis factors such as AA263 reduced intra- and extracellular AAT polymers and improved NE inhibitor activity of secreted AAT in HUH7.5$^{AAT Null}$ cells transiently expressing mutant Z variant AAT (*Sun et al., 2023*). Here, we compared the potential for AA263 and improved analogs of AA263 including AA263$^{yne}$ and AA263-20 to mitigate polymer accumulation and reduced activity of the disease-associated Z variant of AAT stably expressed in liver-derived Huh7 cells (Huh7.5Z). Initially, we confirmed that AA263, AA263$^{yne}$, and AA263-20 induced expression of the ATF6 target gene *BiP* in these cells (*Figure 4—figure supplement 1A*). As reported previously, we found that AA263 reduced intracellular and secreted AAT-Z polymers, as measured by ELISA (*Figure 4A, B*). However, we did not observe AA263-dependent enhancement of AAT NE inhibition activity in conditioned media, as measured by NE substrate-based fluorogenic assay (*Figure 4C*; *Bieth et al., 1974*). In contrast, both AA263$^{yne}$ and AA263-20 reduced intracellular and secreted AAT polymers and enhanced AAT-Z NE inhibition in conditioned media (*Figure 4A–C*). This indicates that AA263$^{yne}$ and AA263-20 both show enhanced potential to rescue AAT-Z secretory proteostasis, as compared to AA263.

## Optimized AA263 analogs rescue the surface trafficking and function of an epilepsy-prone GABA$_A$ receptor variant

We next sought to investigate the potential of prioritized AA263 analogs for modulating the expression and functional activity of destabilized gamma-aminobutyric acid type A (GABA$_A$) receptors. GABA$_A$ receptors are pentameric proteins assembled in the ER from eight subunit classes, with the most common composition being two α1 subunits, two β2/β3 subunits, and one γ2 subunit (*Braat and Kooy, 2015*; *Sigel and Steinmann, 2012*). Various variants in these major subunits have been reported to impair the trafficking of GABA$_A$ receptors to the plasma membrane, ultimately leading to a significant decrease in their function (*Benarroch, 2007*; *Wang et al., 2024*). As such, GABA$_A$ variants have been widely associated with the onset and pathogenesis of numerous neurological disorders, including genetic epilepsy (*Fu et al., 2022*; *Benarroch, 2007*; *Todd et al., 2014*). It has previously been reported that enhancement of ER proteostasis can stabilize GABA$_A$ variants and restore their surface trafficking as well as their function (*Wang et al., 2022a*; *Di et al., 2021*). Thus, we aimed to determine if our optimized AA263 analogs could similarly rescue the surface expression and activity of a known trafficking-deficient γ2(R177G) GABA$_A$ variant, associated with complex febrile seizures (*Todd et al., 2014*).

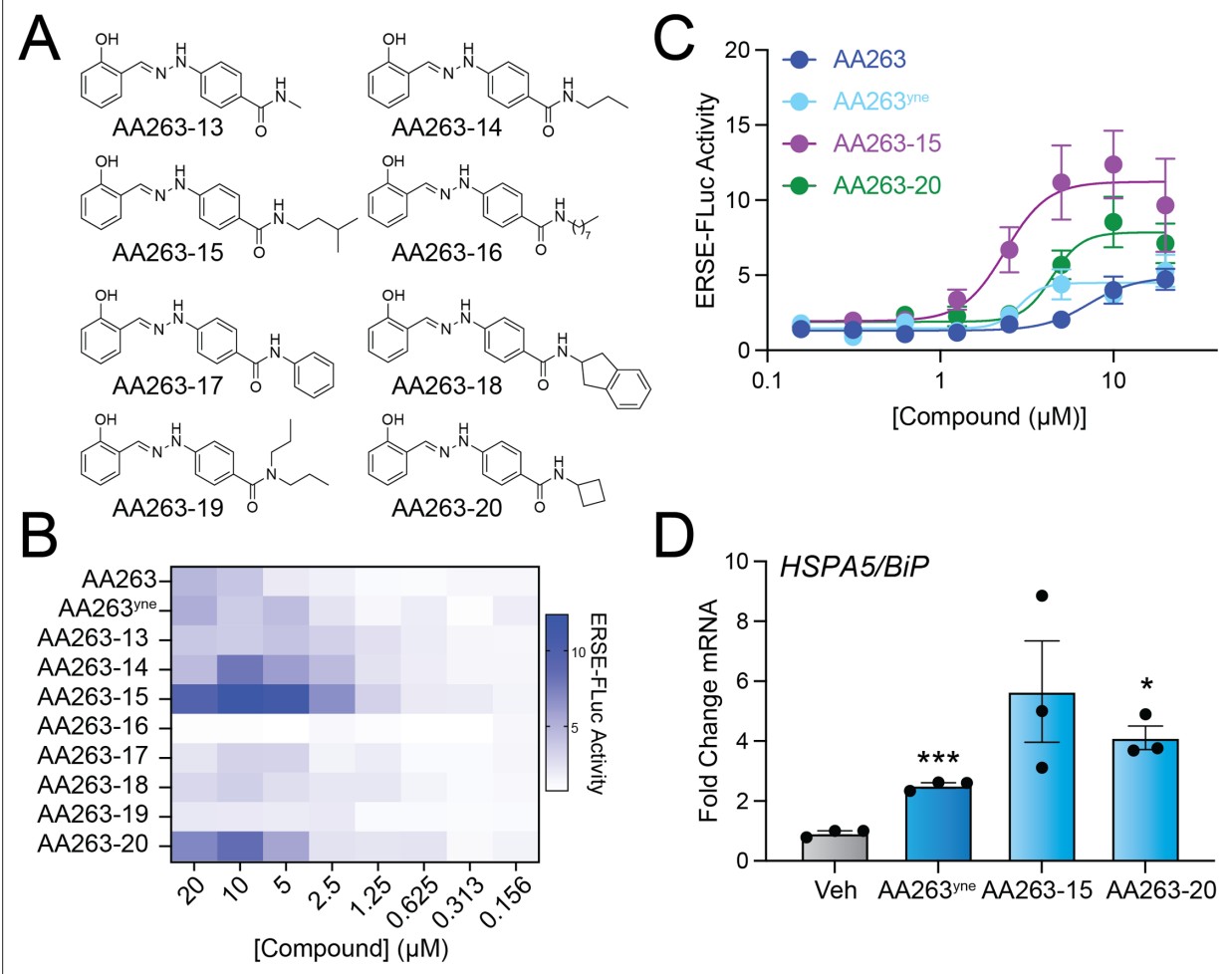

**Figure 3.** Diversification of the AA263 B-ring affords improved AA263 analogs. (**A**) Structures of AA263 analogs. (**B**) Heat map showing activation of the ERSE-FLuc ATF6 reporter in HEK293T cells treated for 18 hr with the indicated dose of compound. (**C**) Activation of the ERSE.Fluc ATF6 reporter in HEK293T cells treated for 18 hr with the indicated dose of compound. Error bars show SEM for $n = 6$ replicates. (**D**) Expression, measured by qPCR, of the ATF6 target gene *BiP* in HEK293T cells treated with indicated AA263 analog (10 µM) for 6 hr. Error bars show SEM for $n = 3$ independent biological replicates. *$p < 0.05$, ***$p < 0.001$ for one-way ANOVA. See also *Figure 3—source data 1*.

The online version of this article includes the following source data and figure supplement(s) for figure 3:

**Source data 1.** Excel spreadsheet showing raw data used to generate *Figure 3B–D*.

**Figure supplement 1.** Diversification of the AA263 B-ring affords improved AA263 analogs.

**Figure supplement 1—source data 1.** Excel spreadsheet showing raw data used to generate *Figure 3—figure supplement 1A*.

**Figure supplement 1—source data 2.** Uncropped gel shown in *Figure 3—figure supplement 1B*.

**Figure supplement 1—source data 3.** Uncropped gel shown in *Figure 3—figure supplement 1B* with the same annotations shown in *Figure 3—figure supplement 1B*.

We treated HEK293T cells stably expressing α1β2γ2(R177G) pathogenic receptors with each AA263 analog. While both AA263[yne] and AA263-5 increased total protein levels of γ2(R177G) (*Figure 5—figure supplement 1A*), only AA263[yne] could increase the surface expression of α1β2γ2(R177G) GABA$_A$ receptors transiently transfected in HEK293T cells (*Figure 5A*). Thus, the optimized compound AA263[yne] effectively rescues the trafficking of this GABA$_A$ variant to the cell surface. Importantly, the increased total protein levels of γ2(R177G) afforded by treatment with AA263[yne] could be partially attenuated by co-treatment with the selective ATF6 inhibitor Ceapin-A7 (CP7) (*Torres et al., 2019*; *Gallagher and Walter, 2016*; *Gallagher et al., 2016*), indicating that compound-dependent ATF6 activation contributes to this observed increase (*Figure 5—figure supplement 1B*). Next, we used cycloheximide (CHX) chase assays to determine the impact of AA263[yne] on the turnover of the

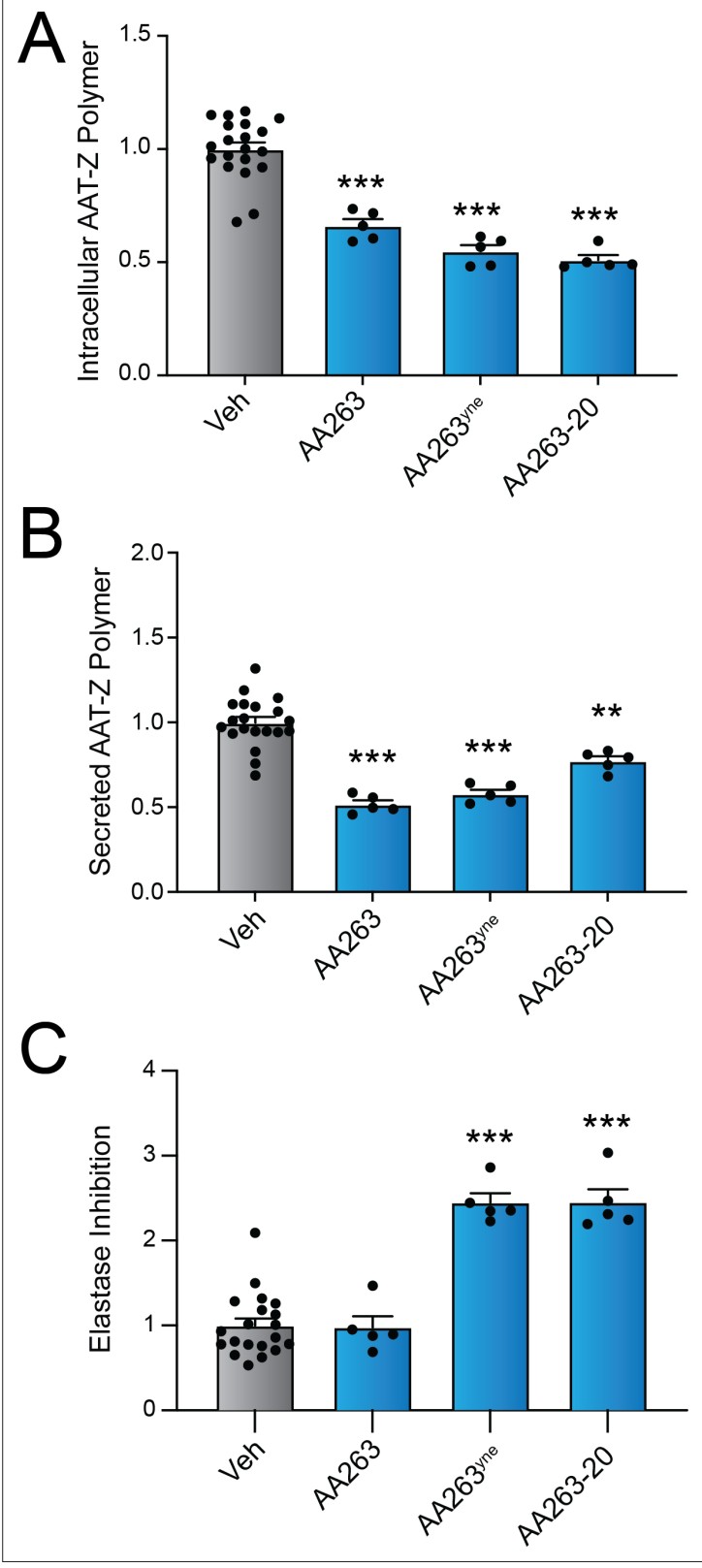

**Figure 4.** AA263 analogs improve secretory proteostasis for the disease-associated AAT-Z variant. Intracellular AAT-Z polymer levels (**A**), extracellular AAT-Z polymer levels in conditioned media (**B**), and elastase inhibition activity of AAT-Z in conditioned media (**C**) from Huh7.5Z cells treated for 24 hr with AA263 (10 μM), AA263[yne] (10 μM), or AA263-20 (10 μM). Error bars show SEM for $n > 5$ replicates. Data are shown normalized to vehicle-

*Figure 4 continued on next page*

*Figure 4 continued*

treated cells. *p < 0.05, **p < 0.01, ***p < 0.005 for one-way ANOVA compared to vehicle-treated cells. See also ***Figure 4—source data 1***.

The online version of this article includes the following source data and figure supplement(s) for figure 4:

**Source data 1.** Excel spreadsheet showing raw data used to generate ***Figure 4A–C***.

**Figure supplement 1.** AA263 analogs improve secretory proteostasis for the disease-associated AAT-Z variant.

**Figure supplement 1—source data 1.** Excel spreadsheet showing raw data used to generate ***Figure 4—figure supplement 1***.

γ2(R177G) variant. Compared with WT receptors, we found that the pathogenic γ2(R177G) variant displayed a faster turnover, which was slowed down by treatment with AA263$^{yne}$ to a similar rate compared to GABA$_A$ WT (***Figure 5—figure supplement 1C***). In addition to AA263$^{yne}$, both AA263-15 and AA263-20 increased the total expression levels of γ2(R177G) subunit in HEK293T cells treated with either compound (***Figure 5—figure supplement 1D***). Further, treatment with these compounds increased surface expression levels of the γ2(R177G) subunit (***Figure 5B***). Together, these results show that AA263$^{yne}$ and optimized analogs such as AA263-20 can efficiently correct the misfolding of an epilepsy-associated GABA$_A$ variant and rescue its trafficking to the plasma membrane.

Finally, to determine if compound-dependent increases of α1β2γ2(R177G) receptor surface expression also translate into a rescued function of the receptor, we used whole-cell patch clamp in HEK293T cells transiently transfected with either WT GABA$_A$ receptor or the α1β2γ2(R177G) GABA$_A$ variant and recorded evoked inhibitory postsynaptic currents (eIPSCs) by a 4-s application of 10 mM GABA. Consistent with the literature, we first showed that cells expressing the epilepsy-associated γ2(R177G) variant displayed eIPSCs of significantly smaller amplitude than cells expressing WT GABA$_A$ receptors (***Figure 5C, D***; ***Wang et al., 2022a***). In addition, after normalizing each cell's amplitude by its capacitance, γ2(R177G) GABA$_A$-containing cells had a lower peak current density when compared with WT GABA$_A$-containing cells (***Figure 5C, E***). Importantly, treatment with AA263$^{yne}$ or AA263-20 in HEK293T cells transiently expressing α1β2γ2(R177G) GABA$_A$ receptors restored both eIPSC peak amplitude and peak current density to levels nearly identical to those observed in cells expressing WT GABA$_A$ receptors (***Figure 5C–E***). Collectively, these findings establish AA263$^{yne}$ and AA263-20 as ER proteostasis regulators that not only restore surface trafficking of disease-associated GABA$_A$ receptor variants, but also rescue their functional activity at the cell surface.

## Discussion

Herein, we define the molecular basis for AA263-dependent ER proteostasis remodeling. We show that AA263 covalently targets a subset of ER-localized PDIs, suggesting that this compound, like AA147, promotes activation of the adaptive ATF6 signaling arm of the UPR through a mechanism involving PDI modification. Intriguingly, unlike AA147, structure–activity relationship studies showed that AA263 activity could be enhanced by altering the putative B-ring of this compound. This identified two analogs, AA263-5 and AA263$^{yne}$, that exhibited enhanced selectivity and efficacy for ATF6 activation. Further optimization of the amide moiety of AA263$^{yne}$ afforded prioritized analogs including AA263-20. We then showed AA263$^{yne}$, and this prioritized analog exhibited enhanced activity relative to the parent compound AA263 in correcting ER proteostasis in cellular models of alpha-1-antitrypsin deficiency and GABA$_A$ receptor channelopathies. These studies demonstrate the potential to enhance ER proteostasis remodeling activity with compound AA263 to correct proteostasis deficits in etiologically diverse protein misfolding diseases.

Unlike other ER proteostasis activators, such as AA147 and AA132, AA263 does not possess the 2-amino-*p*-cresol moiety required for the enzymatic activation of these other compounds to a reactive quinone methide. Despite this, we show that AA263 functions through the covalent targeting of ER-localized PDIs. Salicaldehyde *N*-acylhydrazones can generate transient quinone-methide type species driven by tautomerization (***Ifa et al., 2000***). This suggests that AA263 tautomerization to a quinone methide may underlie this covalent PDI targeting. Consistent with this, we show that AA263 can covalently modify proteins ex vivo in cellular lysates. However, a key question is how does AA263 show selectivity for ER PDIs independent of the localized metabolic activation afforded by other

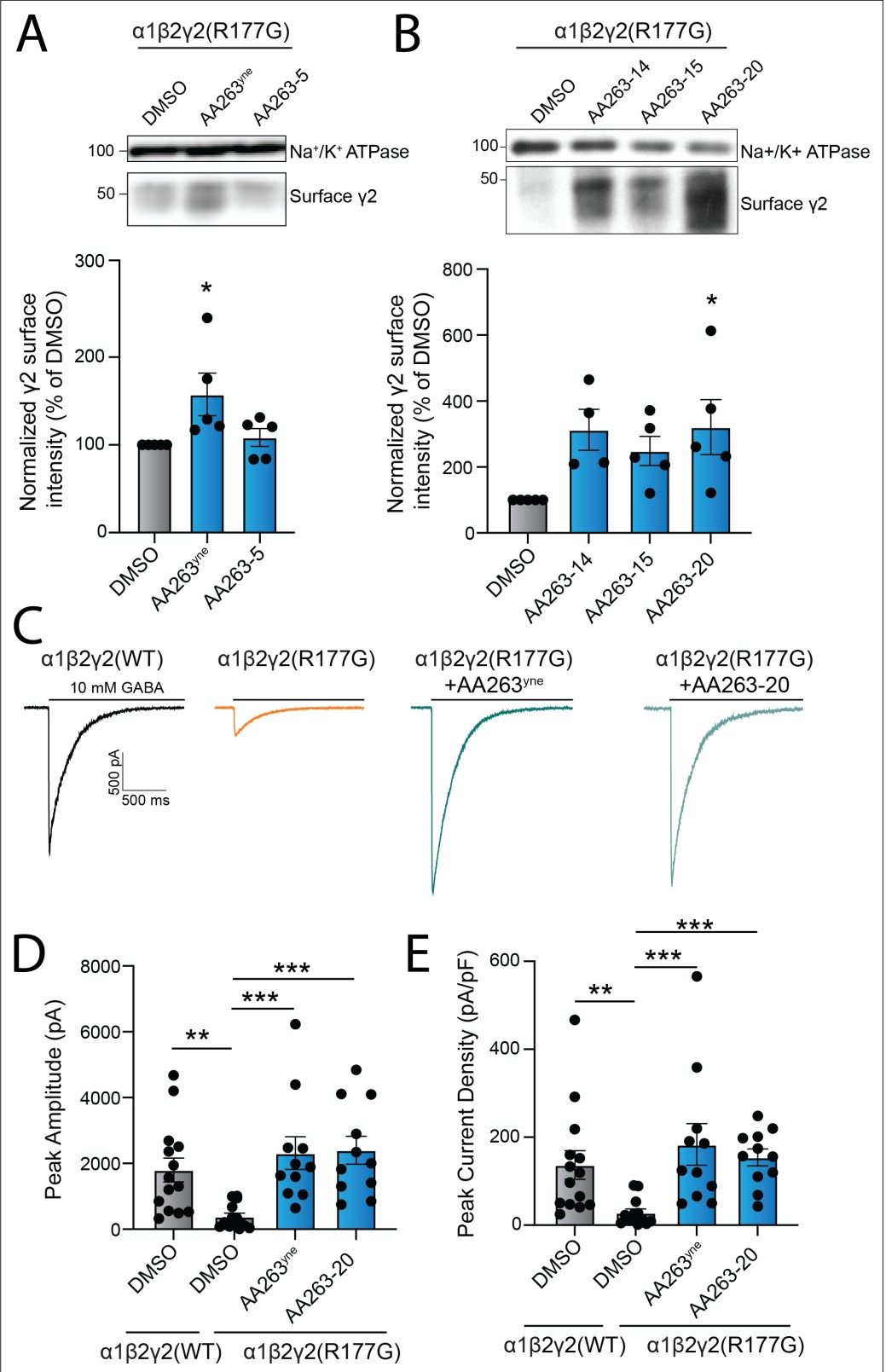

**Figure 5.** Enhanced AA263 analogs promote the trafficking and plasma membrane activity of destabilized, disease-associated GABA$_A$ receptors. (**A**) Representative immunoblot (above) and quantification (below) of surface biotinylated γ2 in HEK293T cells stably expressing α1β2γ2(R177G) GABA$_A$ receptors treated for 24 hr with 10 μM AA263$^{yne}$ or AA263-5. Na$^+$/K$^+$ ATPase serves as a loading control. (**B**) Representative immunoblot showing surface

*Figure 5 continued on next page*

*Figure 5 continued*

γ2 expression in HEK293T cells transiently transfected with α1β2γ2(R177G) receptors and treated with indicated AA263 analogs 10 μM, 24 hr. $Na^+/K^+$ ATPase serves as a loading control. (**C**) Representative evoked inhibitory postsynaptic current (eIPSC) traces for α1β2γ2(WT) $GABA_A$ receptors and α1β2γ2(R177G) $GABA_A$ receptors treated for 24 hr with DMSO, $AA263^{yne}$ (10 μM), or AA263-20 (10 μM). 10 mM GABA (saturating condition) was applied to the recorded cells to evoke currents. Histograms showing changes in eIPSC peak amplitude (**D**) and peak current density (**E**) for the indicated groups. Band intensities were quantified using ImageJ software, normalized to the DMSO control condition. Each data point is reported as mean ± SEM. One-way ANOVA followed by post hoc Dunnett's test was used for statistical analysis for A and B. Kruskal–Wallis test followed by post hoc Dunn's test was used for statistical analysis for D and E. *$p < 0.05$; **$p < 0.01$; ***$p < 0.001$. See also *Figure 5—source data 1* and *Figure 5—source data 2*.

The online version of this article includes the following source data and figure supplement(s) for figure 5:

**Source data 1.** Excel spreadsheet showing raw data used to generate *Figure 5A, B, D, E*.

**Source data 2.** Uncropped gels shown in *Figure 5A and B* with the same annotations shown in *Figure 5A and B*.

**Source data 3.** Uncropped gels shown in *Figure 5A and B*.

**Figure supplement 1.** Enhanced AA263 analogs promote the trafficking and plasma membrane activity of destabilized, disease-associated $GABA_A$ receptors.

**Figure supplement 1—source data 1.** Excel spreadsheet showing raw data used to generate *Figure 5—figure supplement 1A, B, and D*.

**Figure supplement 1—source data 2.** Uncropped gels shown in *Figure 5—figure supplement 1A–D* with the same annotations shown in *Figure 5—figure supplement 1A–D*.

**Figure supplement 1—source data 3.** Uncropped gels shown in *Figure 5—figure supplement 1A–D*.

---

compounds such as AA147. One potential explanation is that the AA263 electrophilic species may possess differential reactivity toward a subset of proteinogenic thiols such as those in PDIs versus glutathione (*Zambaldo et al., 2020*; *Shindo et al., 2019*) Regardless, these results, when combined with our analyses for AA147 and AA132 (*Kline et al., 2023*), demonstrate how the diverse covalent engagement profiles of subtly different electrophilic species against a subset of ER proteins can elicit a spectrum of transcriptional outcomes.

Our findings that AA263 engages similar PDI family members and ER proteins as AA147 further suggest the importance of this family of proteins for adjusting ER proteostasis through mechanisms including activation of ER stress responsive signaling pathways (*Higa et al., 2014*; *Oka et al., 2019*; *Eletto et al., 2014*; *Kranz et al., 2017*). We previously showed that the extent of PDI labeling directly impacts the ability for this class of compounds to activate ATF6, with more modest levels leading to preferential ATF6 activation while higher levels of labeling activating all three arms of the UPR (*Kline et al., 2023*). Interestingly, our results indicate that AA263 maintains preferential selectivity for ATF6 activation despite modifying an intermediate population of PDIs greater than those observed for the selective ATF6 activator AA147, but less than those observed for the global UPR activator AA132. This suggests that higher levels of PDI engagement than those observed with AA147 are still accessible without compromising the ability for these compounds to preferentially activate ATF6.

Intriguingly, like AA147 and AA132, we show that the AA263 scaffold is amenable to modifications on the putative non-reactive B-ring. However, unlike these other compounds, medicinal chemistry efforts focused on improving AA263 activity identified analogs with improved potency and transcriptional selectivity for ATF6 activation relative to the parent compound. The improvements in transcriptional selectivity of the compounds may be attributed to the removal of the nitro group from the parent compound AA263. Nitroarene-containing compounds have been shown to affect mitochondrial function and also lead to ROS production through nitroarene metabolism (*Coe et al., 2007*; *Nepali et al., 2019*). Thus, the crosstalk between ISR signaling and the mitochondria may explain the higher levels of PERK/ISR signaling with AA263 relative to our prioritized analogs (*Bassot et al., 2023*; *Lebeau et al., 2018*; *Verfaillie et al., 2012*). Additionally, we do observe some basal CHOP signaling with inactive AA263 analogs, exemplified by AA263-2, perhaps suggesting some inherent promiscuity of the phenylhydrazone scaffold. As a result, the identity of the B-ring electron-withdrawing group may improve the spectrum of target protein engagement in a positive manner to boost selectivity for ATF6 activation (*Zambaldo et al., 2020*).

It is important to note that both 2-hydroxyphenylhydrazones and 4-hydroxyphenylhydrazones have been classified as pan-assay interference compounds (PAINS), due, in part, to their reactivity toward biological nucleophiles and metal chelation properties (*Baell and Holloway, 2010*; *Baell and Nissink, 2018*; *Baell and Holloway, 2010*; *Pouliot and Jeanmart, 2016*). However, our observation of SAR trends connecting scaffold modifications to magnitude of pathway activation indicates that this particular compound activity is perhaps independent of PAINS activity. Further, many of our negative analogs possess a functional A-ring, and thus would still be expected to possess covalent reactivity if PAINS behavior mediated the activation of stress-responsive signaling (*Park et al., 2011*). Thus, any nonspecific covalent modification, if it exists, seems to be functionally silent in this assay. Moreover, we found that these compounds showed remarkable specificity for ER-localized protein engagement, further indicating that AA263 and prioritized analogs possess other potential properties that drive specificity toward the ER, which we are continuing to pursue.

Optimized AA263 analogs identified herein have enhanced potential to correct ER proteostasis in two distinct cellular models of protein misfolding diseases – AAT deficiency and $GABA_A$ receptor trafficking. We show that AA263 analogs reduce both intracellular and extracellular accumulation of AAT-Z aggregates and improve activity of the secreted protein, correcting both aspects of disease pathology. Further, we demonstrate that our optimized AA263 analogs enhanced trafficking, surface expression, and membrane activity of a pathogenic $GABA_A$ receptor variant associated with genetic epilepsy, restoring its functional activity at the plasma membrane. AA263 and its related analogs can influence ER proteostasis in these models through different mechanisms including ATF6-dependent remodeling of ER proteostasis and direct alterations to the activity of specific PDIs. Consistent with this, we show that pharmacologic inhibition of ATF6 only partially blocks increases of γ2(R177G) afforded by treatment with AA263$^{yne}$, highlighting the benefit for targeting multiple aspects of ER proteostasis to enhance ER proteostasis of this disease-relevant $GABA_A$ variant. While additional studies are required to further deconvolute the relative contributions of these two mechanisms on the protection afforded by our optimized compounds, our results demonstrate the potential for these compounds to enhance ER proteostasis in the context of different protein misfolding diseases. As we and others continue expanding on this class of ER proteostasis regulators, we will further demonstrate the therapeutic utility of these compounds in cellular and in vivo models and identify next-generation compounds with improved potential for translation to correct ER proteostasis defects in etiologically diverse diseases.

## Materials and methods

**Key resources table**

| Reagent type (species) or resource | Designation | Source or reference | Identifiers | Additional information |
|---|---|---|---|---|
| Cell line (human) | HEK293-Trex overexpressing ERSE.Fluc | *Plate et al., 2016* | | |
| Cell line (human) | HEK293-Trex overexpressing XBP1.RLuc | *Plate et al., 2016* | | |
| Cell line (human) | HEK293T overexpressing ATF4.FLuc | *Yang et al., 2023* | | |
| Cell line (human) | HEK293T | ATCC | | |
| Cell line (human) | HEK293T cells stably expressing α1β2γ2(R177G) $GABA_A$ receptors | This manuscript | | See Materials and methods |
| Cell line (human) | Huh7.5 cells stably expressing AAT-Z | *Lu et al., 2022* | | |
| Antibody | Rabbit anti-$GABA_A$R-γ2 polyclonal antibody (#AB5559) | Millipore | RRID:AB_11211236 | 1:1000 |
| Antibody | Rabbit monoclonal anti-Na$^+$/K$^+$-ATPase (#ab76020) | Abcam | RRID:AB_1310695 | 1:10,000 |
| Antibody | Mouse anti-human AAT monoclonal antibody 2C1 (Cat #HM2289) | Hycult Biotech | | 1:1000 |
| Antibody | Mouse anti-human AAT monomer-specific monoclonal antibody 16F8 | Balch Lab (Scripps Research) | | 1:1000 |
| Antibody | Rhodamine anti-actin primary antibody (#12004163) | Bio-Rad | RRID:AB_2861334 | 1:8000 |
| Recombinant DNA reagent | pCMV6 plasmids containing human $GABA_A$ receptor α1 | Origene | Uniprot No. P14867-1 | |

*Continued on next page*

*Continued*

| Reagent type (species) or resource | Designation | Source or reference | Identifiers | Additional information |
|---|---|---|---|---|
| Recombinant DNA reagent | pCMV6 plasmids containing human GABA$_A$ receptor β2 (isoform 2) | Origene | Uniprot No. P47870-1 | |
| Recombinant DNA reagent | pCMV6 plasmids containing human GABA$_A$ receptor γ2 (isoform 2) subunits | Origene | Uniprot No. P18507-2 | |
| Recombinant DNA reagent | pCMV6 encoding human GABA$_A$ receptor γ2 subunit missense mutation R177G | Constructed using QuikChange II site-directed mutagenesis Kit (Agilent Genomics) | | |
| Peptide, recombinant protein | Human neutrophil elastase (Cat # IHUELASD100UG) | Innovation Research | | |
| Commercial assay or kit | QuikChange II site-directed mutagenesis Kit (#200523) | Agilent Genomics | | |
| Commercial assay or kit | Firefly luciferase assay reagent-1 | Targeting Systems | | |
| Commercial assay or kit | Renilla luciferase assay reagent-1 | Targeting Systems | | |
| Commercial assay or kit | QuickRNA Miniprep Kit (R1055) | Zymo | | |
| Commercial assay or kit | High-Capacity cDNA Reverse Transcription Kit | Applied Biosystems | | |
| Commercial assay or kit | PowerSYBR Green PCR Master Mix | Applied Biosystems | | |
| Commercial assay or kit | Micro BCA protein assay (#23235) | Thermo Fisher | | |
| Commercial assay or kit | Tandem Mass Tag (TMT) 10plex (Cat #90110) | Thermo Scientific | | |
| Chemical compound, drug | Cycloheximide (Cat #01810) | Sigma-Aldrich | | |
| Chemical compound, drug | γ-Aminobutyric acid (GABA) (#A2129) | Sigma-Aldrich | | |
| Chemical compound, drug | MG-132 (#A2585) | ApexBio | | |
| Chemical compound, drug | Thapsigargin | Sigma-Aldrich | | |
| Chemical compound, drug | AA263 and AA263 analogs | Synthesized in this manuscript | | See *Supplementary file 3* |
| Chemical compound, drug | Protease inhibitor cocktail (#4693159001) | Roche, Indianapolis, IN | | |
| Chemical compound, drug | Cy5-azide | Click Chemistry Tools, Scottsdale, AZ | | |
| Chemical compound, drug | BTTAA ligand (2-(4-((bis((1-tert-butyl-1$H$-1,2,3-triazol-4-yl)methyl)amino)methyl)-1$H$-1,2,3-triazol-1-yl)acetic acid) | Albert Einstein College | | |
| Chemical compound, drug | Elastase substrate (Z-Ala4)2Rh110 (Cat No. 11675) | Cayman Chemical | | |
| Chemical compound, drug | Sulfo-NHS SS-Biotin (#A8005) | ApexBIO | | |

## Reagents and plasmids

Cycloheximide (Cat #01810) and γ-aminobutyric acid (#A2129) were obtained from Sigma-Aldrich. Human NE (Cat #IHUELASD100UG) was obtained from Innovation Research. MG-132 (#A2585) was obtained from ApexBio. Thapsigargin was purchased from Sigma-Aldrich. AA263 analogs and AA263$^{yne}$ analogs were synthesized as described in Appendix 1 and used as 10 mM DMSO stocks. The pCMV6 plasmids containing human GABA$_A$ receptor α1 (Uniprot No. P14867-1), β2 (isoform 2, Uniprot No. P47870-1), γ2 (isoform 2, Uniprot No. P18507-2) subunits, and pCMV6 Entry Vector plasmid (pCMV6-EV) were obtained from Origene. The human GABA$_A$ receptor γ2 subunit missense mutation R177G was constructed using QuikChange II site-directed mutagenesis Kit (Agilent Genomics). All cDNA sequences were confirmed by DNA sequencing.

## Antibodies

The rabbit anti-GABA$_A$R-γ2 polyclonal antibody (#AB5559) was obtained from Millipore. The rabbit monoclonal anti-Na$^+$/K$^+$-ATPase (#ab76020) antibody was obtained from Abcam. The rhodamine anti-actin primary antibody (#12004163) was obtained from Bio-Rad. Mouse anti-human AAT monoclonal antibody 2C1 was obtained from Hycult Biotech (Cat #HM2289). Mouse anti-human AAT monomer-specific monoclonal antibody 16F8 was a kind gift from the Balch Lab. The secondary goat anti-rabbit

antibody (#A27036) and goat anti-mouse antibody (#A28177) used for western blot were obtained from Invitrogen. The secondary goat anti-mouse HRP antibody used for the AAT ELISA experiment was obtained from Thermo Fisher Scientific (Cat #32230).

## Cell culture

HEK293T-Rex and HEK293T (ATCC) cells were cultured in high-glucose Dulbecco's modified Eagle's medium supplemented with glutamine, penicillin/streptomycin, and 10% fetal bovine serum. Cells were routinely tested for mycoplasma every 6 months. We did not further authenticate the cell lines. All cells were cultured under typical tissue culture conditions (37°C, 5% $CO_2$).

## Generation of stable HEK293T α1β2γ2(R177G) cell line

HEK293T cells were grown and allowed to reach ~70% confluency before transient transfection using TransIT-2020 (Mirus Bio #MIR 5406, Madison, WI) according to the manufacturer's instruction. Stable cell lines for α1β2γ2(R177G) were generated using the antibiotic G-418 selection method. Briefly, cells were transfected with α1β2γ2(R177G) (0.25:0.25:0.5 μg) plasmids, and then selected with 0.8 mg/ml G-418 (Fisher # 50-153-2785) for 7 days. Cells were diluted and passed to 96-well plates to ensure single-cell distribution per well. Cells expanded from monoclonal cells were assessed for the expression of α1, β2, and γ2 by western blot analysis. Positive monoclonal cells were selected for experimentations.

## Measurement of UPR activity using luciferase reporters

HEK293T-Rex cells expressing the ERSE.FLuc (*Plate et al., 2016*), XBP1s.RLuc (*Grandjean and Wiseman, 2020*), or ATF4.FLuc reporter were plated at 80 μl/well from suspensions of 250,000 cells/ml in white clear-bottom 96-well plates (Corning) and incubated at 37°C overnight. The following day, cells were treated with 20 μl of compound-containing media to give final concentration as described before incubating for 18 hr at 37°C. The plates were equilibrated to room temperature, then either 100 μl of Firefly luciferase assay reagent-1 (ERSE.FLuc and ATF4.FLuc) or Renilla luciferase assay reagent-1 (XBP1s.RLuc) (Targeting Systems) were added to each well. Samples were dark adapted for 10 min to stabilize signals. Luminescence was then measured in an Infinite F200 PRO plate reader (Tecan) and corrected for background signal (integration time 500ms). All measurements were performed in biologic triplicate.

## Quantitative RT-PCR

The relative mRNA expression levels of target genes were measured using quantitative RT-PCR. Cells were treated as described at 37°C, harvested by trypsinization, washed with Dulbecco's phosphate-buffered saline (PBS, Gibco), and then RNA was extracted using the QuickRNA Miniprep Kit (Zymo). qPCR reactions were performed on cDNA prepared from 500 ng of total cellular RNA using the High-Capacity cDNA Reverse Transcription Kit (Applied Biosystems). PowerSYBR Green PCR Master Mix (Applied Biosystems), cDNA, and appropriate primers purchased from Integrated DNA Technologies (see Table below) were used for amplifications (6 min at 95°C, then 45 cycles of 10 s at 95°C, 30 s at 60°C) in an ABI 7900HT Fast Real Time PCR machine. Primer integrity was assessed by a thermal melt to confirm homogeneity and the absence of primer dimers. Transcripts were normalized to the housekeeping genes RPLP2, and all measurements were performed in biological triplicate. Data were analyzed using the RQ Manager and DataAssist 2.0 software (ABI). qPCR data are reported as mean ± standard deviation plotted using Prism GraphPad.

## Sequences of primers for qPCR

| Gene | Forward primer | Reverse primer |
| --- | --- | --- |
| *HSPA5 (BIP)* | GCCTGTATTTCTAGACCTGCC | TTCATCTTGCCAGCCAGTTG |
| *ERDJ4* | GGAAGGAGGAGCGCTAGGTC | ATCCTGCACCCTCCGACTAC |
| *CHOP* | ACCAAGGGAGAACCAGGAAACG | TCACCATTCGGTCAATCAGAGC |
| *RPLP2* | CGTCGCCTCCTACCTGCT | CATTCAGCTCACTGATAACCTTG |

## SDS–PAGE in-gel fluorescence scanning

Indicated cell line (250,000 cells/well) was treated with indicated compound or combination of compounds in 6-well plates at 10 µM for 6 hr. Cells were lysed in radioimmunoprecipitation assay (RIPA) buffer (150 mM NaCl, 50 mM Tris pH 7.5, 1% Triton X-100, 0.5% sodium deoxycholate, and 0.1% SDS) supplemented with fresh protease inhibitor cocktail (Roche, Indianapolis, IN) and centrifuged for 10 min at 16,000 × $g$ following a 30-min incubation. Protein concentration of supernatant was determined by BCA assay (Thermo Fisher) and normalized to give 42.5 µl at 2.35 mg/ml (100 µg/total protein). 7.5 µl 'click chemistry master mix' was added to each sample to give final concentrations of 100 µM of Cy5-azide (Click Chemistry Tools, Scottsdale, AZ), 800 µM copper (II) sulfate, 1.6 mM BTTAA ligand (2-(4-((bis((1-tert-butyl-1H-1,2,3-triazol-4-yl)methyl)amino)methyl)-1$H$-1,2,3-triazol-1-yl) acetic acid) (Albert Einstein College), and 5 mM sodium ascorbate. Reaction incubated at 30°C for 1 hr while shaking before CHCl$_3$/MeOH protein precipitation. Dried protein was redissolved in 20 µl 1X SDS loading buffer and 25 µg was loaded on gel for SDS–PAGE in-gel fluorescence scanning and subsequent visualization using an Odyssey Infrared Imaging System (Li-Cor Biosciences) or Bio-Rad gel imager.

## Comparative AA132 versus AA263 chemoproteomic experiments

HEK293T cells in 10 cm plates at 80–90% confluency were treated for 6 hr with vehicle (0.1% DMSO), AA132yne (10 µM), or AA263-5 (10 µM) at 37°C. The cells were washed with PBS before harvesting with trypsin, pelleting (500 × $g$, 5 min), and washing with PBS (1 ml). Cell pellets were resuspended in RIPA buffer before sonication with a probe tip sonicator for cell lysis (15 s, 3 s on/2 off, 30% amplitude). For each sample, 1 g lysate (500 µl) was reacted with click reagents to give final concentrations as follows: 100 µM of diazo biotin-azide (Click Chemistry Tools, Scottsdale, AZ), 800 µM copper (II) sulfate, 1.6 mM BTTAA ligand (2-(4-((bis((1-tert-butyl-1$H$-1,2,3-triazol-4-yl)methyl)amino)methyl)-1$H$-1,2,3-triazol-1-yl)acetic acid) (Albert Einstein College), and 5 mM sodium ascorbate. The reaction was placed on a shaker at 1000 rpm at 30°C for 90 The reaction was quenched with the sequential addition of cold methanol (4x volume), chloroform (1 x volume), and DPBS (4 x volume) to precipitate proteins. Proteins were pelleted by centrifugation (4700 × $g$, 10 min, 4°C). The supernatant was discarded, and the pellets dried under air for 5 min. Protein pellets were resuspended in 6 M urea in PBS (500 µl) with brief sonication. 50 µl of high-capacity streptavidin beads were washed with PBS and mixed with the protein solution in 6 ml of PBS. This suspension was placed on a rotator or a shaker and agitated for 2 hr. The beads were centrifuged and washed five times with PBS and 1% SDS. The protein was eluted from the beads by two washes of 50 mM sodium dithionite in 1% SDS for 1 hr and then precipitated by chloroform/methanol precipitation as described above. 50 µl of freshly made 1:1 mixture 200 mM TCEP·HCl in DPBS and 600 mM K$_2$CO$_3$ in DPBS was added to each sample before incubation at 37°C for 30 min while shaking. Alkylation of reduced thiols was achieved by addition of 70 µl freshly prepared 400 mM iodoacetamide in DPBS and incubation at room temperature while protected from light. The reaction was quenched by adding 130 µl of 10% SDS in DPBS and then diluted to approximately 0.2% SDS via DPBS (5.5 ml) and incubated with preequilibrated streptavidin agarose beads (3 × 1 ml PBS wash). The samples were rotated at room temperature for 1.5 hr, centrifuged at 2000 rpm for 2 min, and then washed sequentially with 5 ml 0.2% SDS in DPBS, 5 ml DPBS, and 5 ml 100 mM TEAB (Thermo Cat #90114) pH 8.5 to remove non-binding proteins. The beads were transferred to low-bind 1.5 ml Eppendorf tubes and the bound proteins digested overnight at 37°C in 200 µl 100 mM TEAB containing 2 µg sequencing grade porcine trypsin, 1 mM CaCl$_2$, and 0.01% ProteaseMax (Promega Cat #V2071). The beads were centrifuged at 2000 rpm for 5 min to separate the beads from the supernatant. 200 µl supernatant was transferred to a new tube using a gel-loading tip, and the beads were washed with 100 µl TEAB buffer. The beads were centrifuged at 2000 rpm for 5 min and the supernatant was combined with the previous. 120 µl acetonitrile was added to each supernatant sample before addition of 80 µl (200 µg) of TMT 10 plex (Thermo Scientific, Cat #90110) reconstituted in acetonitrile. The samples were incubated at room temperature for 1 hr and vortexed occasionally. 7 µl of freshly prepared 5% hydroxylamine in water was added to each sample to quench the reaction, vortexed, and incubated for 15 min before quenching with addition of 5 µl MS-grade formic acid. The samples were then vacuum centrifuged to dryness. The samples were combined by redissolving the contents of one tube in 200 µl 0.1% trifluoroacetic acid solution in water and sequential transfer to the respective

multiplexed experiment until all samples were redissolved. This stepwise process was repeated with an additional 100 μl 0.1% TFA solution for a final volume of 300 μl. The pooled samples were fractionated using the Pierce high pH Reversed-Phase Fractionation Kit (Thermo Fisher Scientific 84868) according to the manufacturer's instructions. The peptide fractions were eluted from the spin column with solutions of 0.1% triethylamine containing an increasing concentration of MeCN (5–95% MeCN; eight fractions). Samples were dried via vacuum centrifugation, reconstituted in 50 μl 0.1% formic acid, and stored at –80°C until ready for mass spectrometry analysis.

LC–MS/MS analysis was performed using a Q Exactive mass spectrometer equipped with an EASY nLC 1000 (Thermo Fisher). The digest was injected directly onto a 30 cm, 75 μm ID column packed with BEH 1.7 μm C18 resin (Waters). Samples were separated at a flow rate of 200 nl/min on a nLC 1000 (Thermo). Buffer A and B were 0.1% formic acid in water and acetonitrile, respectively. A gradient of 5–40% B over 110 min, an increase to 50% B over 10 min, an increase to 90% B over another 10 min and held at 90% B for a final 10 min of washing was used for 140 min total run time. The column was re-equilibrated with 20 μl of buffer A prior to the injection of sample. Peptides were eluted directly from the tip of the column and nanosprayed directly into the mass spectrometer by application of 2.5 kV voltage at the back of the column. The Q Exactive was operated in a data-dependent mode. Eluted peptides were scanned from 400 to 1800 *m/z* with a resolution of 30,000 and the mass spectrometer in a data-dependent acquisition mode. The top 10 peaks for each full scan were fragmented by HCD using a normalized collision energy of 30%, a 100-ms activation time, a resolution of 7500, and scanned from 100 to 1800 *m/z*. Dynamic exclusion parameters were 1 repeat count, 30-ms repeat duration, 500 exclusion list size, 120 s exclusion duration, and exclusion width between 0.51 and 1.51. Peptide identification and protein quantification was performed using the Integrated Proteomics Pipeline Suite (IP2, Integrated Proteomics Applications, Inc, San Diego, CA) as described previously.

## RNAseq

Cells were lysed and total RNA collected using the Quick-RNA Miniprep kit from Zymo Research (R1055) according to the manufacturer's instructions. RNA concentration was then quantified by NanoDrop. Whole transcriptome RNA was then prepared and sequenced by BGI Americas on the BGI Proprietary platform, which provided paired-end 50 bp reads at 20 million reads per sample. Each condition was performed in triplicate. RNAseq reads were aligned using DNAstar Lasergene SeqManPro to the Homo_sapiens-GRCh38.p7 human genome reference assembly, and assembly data were imported into ArrayStar 12.2 with QSeq (DNASTAR Inc) to quantify the gene expression levels and normalization to reads per kilobase per million. Differential expression analysis was assessed using DESeq2 in R, which also calculated statistical significance calculations of treated cells compared to vehicle-treated cells using a standard negative binomial fit of the reads per kilobase per million data to generate fold-change quantifications. The complete RNAseq data is deposited in Gene Expression Omnibus (GEO) as GSE309691.

## Immunoblotting

Cells were grown in 6-well plates or 10 cm dishes and allowed to reach ~70% confluency before transient transfection with α1:β2:γ2(WT) or α1:β2:γ2(R177G) (0.25:0.25:0.5 μg for one well of 6-well plates or 0.8:0.8:1.6 μg for one 10 cm dish) plasmids using TransIT-2020 (# Mirus Bio, Madison, WI) according to the manufacturer's instruction. The mixture was incubated for 15–20 min at room temperature before being added to the cells. After indicated treatment, cells were harvested with Trypsin and lysed with pH 7.5 lysis buffer (50 mM Tris, 150 mM NaCl, 2 mM *n*-dodecyl-b-D-maltoside (DDM) (GoldBio, catalog #: DDM5)) (pH 7.5) supplemented with complete protease inhibitor cocktail (Roche #4693159001). Lysates were cleared by centrifugation (21,000 × *g*, 10 min, 4°C). Protein concentrations were then measured through microBCA assay (Thermo Fisher Pierce #23235). Aliquots of cell lysates were loaded with 4x Laemmli buffer (Bio-Rad #1610747, CA) with 10% 2-Mercaptoethanol (Sigma-Aldrich #M3148, Saint Louis, MO) and separated in an 8% SDS–PAGE gel. The β-actin serves as a total protein loading control, whereas Na$^+$/K$^+$-ATPase serves as a plasma membrane protein loading control. Band intensity was quantified using ImageJ software from the NIH. For both surface and total expression quantification, the protein level was first normalized to the loading control (β-actin or Na$^+$/K$^+$-ATPase) and then to the vehicle control (DMSO or WT).

## AAT-Z specific conformation (monomer or polymer) ELISA assay

Huh7.5Z cells stably expressing AAT-Z (E342K) (*Lu et al., 2022*) were seeded in 96-well plates at a density of $2.5 \times 10^4$ cells per well. Cells were treated with compounds for 24 hr before collecting the cell culture medium. Wells were aspirated of residual media and refilled with 80 µl/well of lysis buffer (50 mM Tris, 150 mM NaCl, 1% (vol/vol) Triton X-100, and 1X Halt protease inhibitor cocktail) and incubated on 4°C shaker for 1 hr before collecting the resultant lysate. The previous day, clear flat bottom polystyrene high binding 96-well plates were pre-coated with goat anti-human AAT polyclonal antibody (A80-122A 1:1000 in coating buffer (28.6 mM g $Na_2CO_3$, 71.4 mM $NaHCO_3$, pH = 9.6)) at 4°C. Coated plates were washed 5x with PBST washing buffer (1X PBS + 0.05% (vol/vol) Tween-20), blocked for 2–4 hr in washing buffer containing 5% dehydrated milk, and washed another 5x with to remove blocking agent just prior to collection of treated cell products. 20 µl of culture medium or lysate from each treated well was dispensed into the antibody-coated plates and incubated overnight (8–12 hr) at 4°C. Plates were washed 5x with PBST washing buffer before applying conformation-specific primary antibodies for AAT monomer (16F8 *Wang et al., 2022b* 1:2000 in 1X PBS) or polymer (2C1 *Miranda et al., 2010* 1:1000 in 1X PBS) and incubating 2–4 hr at room temperature. Plates were washed 5x with PBST washing buffer before applying the secondary HRP-conjugated goat-anti-mouse antibody (1:5000 in 1x PBS) and incubating 2–4 hr at room temperature. After washing a final 5x with PBST washing buffer, TMB substrate (Thermo Fisher Scientific, No. 34029) was added for 10 min before quenching the reaction with 2 M sulfuric acid. Endpoint ELISA signal (450 nm absorbance) was quantified by BioTek Synergy H1 Hybrid plate reader. AAT-Z monomer and polymer levels in media and lysate following 24 hr treatment with AA263 analogs were normalized to vehicle treated Huh7.5Z cells.

## AAT-Z inhibitory activity (fluorogenic elastase substrate turnover) assay

Culture medium from treated Huh7.5Z cells was collected and added to pre-coated high binding 96-well plates as described above for overnight (8–12 hr) incubation at 4°C. Plates were washed 5x with PBST washing buffer before dispensing a solution of porcine elastase (PPE, Promega, No. V1891) at 4 ng/well in reaction buffer (50 mM Tris, pH 8.5) and incubating the plate for 1 hr at 37°C. Then, 25 pmol/well of the fluorogenic elastase substrate (Z-Ala4)2Rh110 (Cayman Chemical, No. 11675) was added, and the plate was incubated for 1.5 hr at 37°C. The plate was read by a BioTek Synergy H1 Hybrid plate reader at 485 nm excitation and 525 nm emission. Elastase inhibitory activity of secreted AAT-Z in media following 24 hr treatment with AA263 analogs was normalized to vehicle-treated Huh7.5Z cells.

## Biotinylation of cell surface proteins

HEK293T cells were plated in 6 cm dishes for surface biotinylation assays. Cells were transfected with $GABA_A$ receptor variants as indicated 48 hr prior to harvest. Twenty-four hours post-incubation with the indicated compounds, intact cells were washed with ice-cold Dulbecco's phosphate buffered saline (DPBS, VWR #10128-844) and incubated with the membrane-impermeable biotinylation reagent Sulfo-NHS SS-Biotin (1 mg/ml; ApexBIO, #A8005) in DPBS containing 1 mM $CaCl_2$ and 1 mM $MgCl_2$ (DPBS + CM) for 30 min at 4°C to label surface membrane proteins. Cells were then incubated with 50 mM glycine (pH 7.5) twice at 4°C for 5 min to quench reaction. To block sulfhydryl groups, the cells were then incubated with 5 nM *N*-ethyl-maleimide (NEM) in DPBS for 15 min at room temperature. Cells were then solubilized overnight at 4°C in lysis buffer containing 2 mM DDM, 50 mM Tris-HCl, 150 mM NaCl (pH 7.5) supplemented with Roche complete protease inhibitor cocktail (Roche #04,693,116,001) and 5 mM NEM. To pellet cellular debris, the lysates were cleared by centrifugation (21,000 × *g*, 10 min, 4°C). The supernatant containing the biotinylated surface proteins was affinity-purified by incubating for 1 hr at 4°C with 40 µl of immobilized neutravidin-conjugated agarose bead slurry (VWR #PI29201). The samples were then subjected to centrifugation (21,000 × *g*, 10 min, 4°C). The beads were washed three times with a buffer containing in mM: 50 Tris-HCl, 150 NaCl (pH 7.5). Surface proteins were finally eluted from beads by vortexing for 30 min with 80 µl of a buffer containing 2x Laemmli sample buffer, 100 mM DTT, and 6 M urea (pH 6.8) for SDS–PAGE and western blotting analysis.

## Cycloheximide-chase assay

HEK293T cells transiently transfected with α1β2γ2(WT) $GABA_A$ receptors or stably expressing α1β2γ2(R177G) $GABA_A$ receptors were seeded in 6-well plates and incubated at 37°C overnight.

Cells were then treated with 10 μM compound for 24 hr prior to the commencement of the experiment. To stop protein translation, cells were treated with 100 μg/ml cycloheximide (Enzo Life Sciences #ALX-380-269). Cells were then chased for the indicated times, harvested, and lysed for protein analysis.

## Electrophysiological recordings in transfected HEK cells by whole-cell patch clamp

GABA$_A$ α1β2γ2 (WT or R177G) subunits were transfected 48 hr prior to recordings with α1:β2:γ2 (0.25:0.25:0.5 μg) into HEK293T cells using TransIT-2020 reagent (Mirus Bio #MIR 5406, Madison, WI). The α1(WT) subunit was co-transfected with GFP, which acted as an expression marker. ATF6 activators at 10 μM were added to the cultures 24 hr post-transfection, and 24 hr prior to the commencement of the experiments.

All experiments were performed at room temperature in the whole-cell configuration obtained in voltage-clamp mode at a holding potential of –60 mV. Patch pipettes with a resistance of 3–5 MΩ were made from borosilicate glass (DWK Life Sciences, #34500-99) on a two-step micropipette puller (Narishige PP-830). The intracellular solution contained the following (in mM): 153 KCl, 1 MgCl$_2$, 5 ethylene glycolbis(b-aminoethyl ether)-$N$,$N$,$N9$,$N9$-tetraacetic acid (EGTA), 10 4-(2-hydroxyethyl)-1-piperazineethanesulfonic acid (HEPES), and 5 Mg-ATP with pH adjusted to 7.3 and osmolarity at 310 mOsm. Cells were continuously perfused with an extracellular solution composed of (in mM): 142 NaCl, 8 KCl, 6 MgCl$_2$, 1 CaCl$_2$, 10 glucose, and 10 HEPES with pH adjusted to 7.4 and osmolarity at 290 mOsm.

Each experiment consisted of 1-min recording in extracellular solution with the application of 10 mM GABA (Sigma-Aldrich #A2129, Saint Louis, MO) for 4 s every 6 s to the recorded cell via a theta glass (Siskiyou #15700000E) controlled by a high-speed piezo solution switcher (Siskiyou, MXPZT-300 series).

eIPSCs were recorded using an Axopatch 200B amplifier and pClamp 10 software, filtered at 1 kHz and sampled at 20 kHz. The first action potential of each recording was analyzed for peak amplitude (p$A$) using Clampfit 11.2. Normalization of currents to each cell's capacitance (p$F$) was performed to allow for collection of current density data (p$A$/p$F$). Recordings with an access resistance >12 MΩ were not included in the analysis. For each experiment, GFP fluorescence was used to control for successful expression of the GABA$_A$ receptor in the cells.

## Quantification and statistical analysis

Statistical analyses were performed using GraphPad Prism 10 (GraphPad, San Diego, CA). All data were first tested for a Gaussian distribution using a Shapiro–Wilk test. The statistical significance between means was determined by either parametric or nonparametric analysis of variance followed by post hoc comparisons (Dunn's, Dunnet) using GraphPad Prism 10. Statistical significance was set at a <0.05. Error bars in the graphs represent mean ± SEM.

## Acknowledgements

We thank Leonard Yoon for help in analyzing the RNAseq experiments described in this manuscript. This work was supported by the National Institutes of Health (AG046495 to RLW and JWK; DK137470 to RLW; NS105789 to TM; HL169631 to WEB).

## Additional information

### Competing interests

Jeffery W Kelly: Shareholder and scientific advisory board member of Protego Biopharma, which has licensed ATF6 activators for therapeutic development. Tingwei Mu: Scientific advisory board member of Cure GABA-A Variants Foundation, a nonprofit organization. R Luke Wiseman: Reviewing editor, eLife. The other authors declare that no competing interests exist.

## Funding

| Funder | Grant reference number | Author |
| --- | --- | --- |
| National Institutes of Health | AG046495 | Jeffery W Kelly<br>Luke Wiseman |
| National Institutes of Health | DK137470 | R Luke Wiseman |
| National Institutes of Health | NS105789 | Tingwei Mu |
| National Institutes of Health | HL169631 | William E Balch |

The funders had no role in study design, data collection, and interpretation, or the decision to submit the work for publication.

## Author contributions

Gabriel M Kline, Adrian Guerrero, Conceptualization, Formal analysis, Investigation, Visualization, Writing – original draft, Writing – review and editing; Lisa Boinon, Conceptualization, Formal analysis, Validation, Investigation, Writing – original draft, Writing – review and editing; Sergei Kutseikin, Gabrielle Cruz, Marnie P Williams, Formal analysis, Investigation, Visualization, Writing – review and editing; Ryan J Paxman, Formal analysis, Investigation, Visualization; William E Balch, Funding acquisition, Project administration; Jeffery W Kelly, Funding acquisition, Project administration, Writing – review and editing; Tingwei Mu, Formal analysis, Funding acquisition, Visualization, Writing – review and editing; R Luke Wiseman, Conceptualization, Formal analysis, Funding acquisition, Writing – original draft, Project administration, Writing – review and editing

## Author ORCIDs

William E Balch ⓘ https://orcid.org/0000-0003-0899-8381
Jeffery W Kelly ⓘ https://orcid.org/0000-0001-8943-3395
Tingwei Mu ⓘ https://orcid.org/0000-0002-6419-9296
R Luke Wiseman ⓘ https://orcid.org/0000-0001-9287-6840

Reviewer #1 (Public review): https://doi.org/10.7554/eLife.107000.3.sa1
Reviewer #2 (Public review): https://doi.org/10.7554/eLife.107000.3.sa2
Reviewer #3 (Public review): https://doi.org/10.7554/eLife.107000.3.sa3
Author response https://doi.org/10.7554/eLife.107000.3.sa4

# Additional files

### Supplementary files

Supplementary file 1. RNAseq analysis of HEK293 cells treated with AA263 (10 µM), AA263$^{yne}$ (10 µM), or AA263-5 (10 µM). There are four sheets in this Excel workbook. DESEQ outputs for RNAseq in HEK293 cells treated with 10 µM AA263, AA263$^{yne}$, or AA263-5 for 6 hr and a sheet showing the geneset profiling of different stress-responsive signaling pathways from these RNAseq data.

Supplementary file 2. GO analysis of RNAseq from HEK293 cells treated with AA263 (10 µM), AA263$^{yne}$ (10 µM), or AA263-5 (10 µM). There are three sheets in this Excel workbook showing the GO analysis results of RNAseq data from HEK293 cells treated for 6 hr with 10 µM AA263, AA263$^{yne}$, or AA263-5. GO analysis was performed on genes induced greater than 1.5-fold with an adjusted p value less than 0.05.

Supplementary file 3. Document describing the synthesis and characterization of AA263 analogs discussed in this article.

MDAR checklist

## Data availability

RNA sequence data were deposited in GEO (accession number GSE309691). All data generated or analyzed during this study are included in the manuscript and supporting files; source data files have been provided for all figures.

The following dataset was generated:

| Author(s) | Year | Dataset title | Dataset URL | Database and Identifier |
|---|---|---|---|---|
| Kline GM, Yoon L, Kelly JW, Wiseman RL | 2025 | Phenylhydrazone-based Endoplasmic Reticulum Proteostasis Regulator Compounds with Enhanced Biological Activity | https://www.ncbi.nlm.nih.gov/geo/query/acc.cgi?acc=GSE309691 | NCBI Gene Expression Omnibus, GSE309691 |

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
