## [Editor Report · eLife Assessment]

This study reports the **important** development and characterization of next-generation analogs of the molecule AA263, which was previously identified for its ability to promote adaptive ER proteostasis remodeling. The evidence supporting the conclusions is **convincing**, with rigorous assays used to benchmark the changes in potency and efficacy of the AA263 analogs as well as AA263 targets. The ability of AA263 analogs to restore the loss of function associated with disease-associated proteins prone to misfolding will be of interest to pharmacologists, chemical biologists, and cell biologists, as well as those working on protein misfolding disorders.

---

## [Referee Report · Reviewer #1 (Public review)]

Summary:

This study builds off prior work that focused on the molecule AA147 and its role as an activator of the ATF6 arm of the unfolded protein response. In prior manuscripts, AA147 was shown to enter the ER, covalently modify a subset of protein disulfide isomerases (PDIs), and improve ER quality control for the disease-associated mutants of AAT and GABAA. Unsuccessful attempts to improve the potency of AA147 have led the authors to characterize a second hit from the screen in this study: the phenylhydrazone compound AA263. The focus of this study on enhancing biological activity of the AA147 molecule is compelling, and overcomes a hurdle of the prior AA147 drug that proved difficult to modify. The study successfully identifies PDIs as a shared cellular target of AA263 and its analogs. The authors infer, based on the similar target hits previously characterized for AA147, that PDI modification likely accounts for a mechanism of action for AA263.

Strengths:

The work establishes the ability to modify the AA263 molecule to create analogs with more potency and efficacy for ATF6 activation. The "next generation" analogs are able to enhance the levels of functional AAT and GABAA receptors in cellular models expressing the Z-variant of AAT or an epilepsy-associated variant of the GABAA receptor, outlining the therapeutic potential for this molecule and laying the foundation for future organism-based studies.

The authors are able to establish that like AA147, AA263 covalently targets ER PDIs. While it is a likely mechanism that AA263 works through the PDIs, the authors are careful to discuss that this is a potential mechanism that remains to be explicitly proven. The study provides the foundation for future work to further define a role for the PDIs in the actions of AA263.

---

## [Referee Report · Reviewer #2 (Public review)]

Modulating the UPR by pharmacological targeting of its sensors (or regulators) provides mostly uncharted opportunities in diseases associated with protein misfolding in the secretory pathway. Spearheaded by the Kelly and Wiseman labs, ATF6 modulators were developed in previous years that act on ER PDIs as regulators of ATF6. However, hurdles in their medicinal chemistry have hampered further developments. In this study, the authors provide evidence that the small molecule AA263 also targets and covalently modifies ER PDIs with the effect of activating ATF6. Importantly, AA263 turned out to be amenable to chemical optimization while maintaining its desired activity. Building on this, the authors show that AA263 derivatives can improve aggregation, trafficking and function of two disease-associated mutants of secretory pathway proteins. Together, this study provides compelling evidence for AA263 (and its derivatives) being interesting modulators of ER proteostasis. Mechanistic details of its mode of action will need more attention in future studies that can now build on this.

In detail, the authors provide strong evidence that AA263 covalently binds to ER PDIs, which will inhibit the protein disulfide isomerase activity. ER PDIs regulate ATF6, and thus their finding provides a mechanistic interpretation of AA263 activating the UPR. It should be noted, however, that AA263 shows broad protein labeling (Fig. 1G) which may suggest additional targets, beyond the ones defined as MS hits in this study. Also, a further direct analysis of the IRE1 and PERK pathways (activated or not by AA263) may be an interesting future directions, as e.g. PDIA1, a target of AA263, directly regulates IRE1 (Yu et al., EMBOJ, 2020) and other PDIs also act on PERK and IRE1. The authors interpret modest activation of IRE1/PERK target genes (Fig. 2C) as an effect on target gene overlap, indeed the most likely explanation based on their selective analyses on IRE1 (ERdj4) and PERK (CHOP) downstream genes, but direct activation due to the targeting of their PDI regulators is also a possible explanation. Further key findings of this paper are the observed improvement of AAT behavior and GABAA trafficking and function. Further strength to the mechanistic conclusion that ATF6 activation causes this could be obtained by using ATF6 inhibitors/knockouts in the presence of AA263 (as the target PDIs may directly modulate behavior of AAT and/or GABAA). Along the same line, it also warrants further investigation in future studies why the different compounds, even if all were used at concentrations above their EC50, had different rescuing capacities on the clients.

Together, the study now provides a strong basis for such in-depth mechanistic analyses.

---

## [Referee Report · Reviewer #3 (Public review)]

Summary:

This study aims to develop and characterize phenylhydrazone-based small molecules that selectively activate the ATF6 arm of the unfolded protein response by covalently modifying a subset of ER-resident PDIs. The authors identify AA263 as a lead scaffold and optimize its structure to generate analogs with improved potency and ATF6 selectivity, notably AA263-20. These compounds are shown to restore proteostasis and functional expression of disease-associated misfolded proteins in cellular models involving both secretory (AAT-Z) and membrane (GABAA receptor) proteins. The findings provide valuable chemical tools for modulating ER proteostasis and may serve as promising leads for therapeutic development targeting protein misfolding diseases.

Strengths:

The study presents a well-defined chemical biology framework integrating proteomics, transcriptomics, and disease-relevant functional assays.

Identification and optimization of a new electrophilic scaffold (AA263) that selectively activates ATF6 represents a valuable advance in UPR-targeted pharmacology.

SAR studies are comprehensive and logically drive the development of more potent and selective analogs such as AA263-20.

Functional rescue is demonstrated in two mechanistically distinct disease models of protein misfolding-one involving a secretory protein and the other a membrane protein-underscoring the translational relevance of the approach.

Weaknesses:

ATF6 activation is primarily inferred from reporter assays and transcriptional profiling; direct biochemical evidence of ATF6 cleavage or nuclear translocation remains missing. However, the authors have added supporting data showing that co-treatment with the ATF6 inhibitor CP7 suppresses target gene induction, which partially strengthens the evidence for ATF6-dependent activity.

Although the proposed mechanism involving PDI modification and ATF6 activation is plausible, it is still not experimentally demonstrated and remains incompletely characterized.

In vivo validation is absent, and thus the pharmacological feasibility, selectivity, and bioavailability of these compounds in physiological systems remain untested.

Comments on revisions:

The authors have generally addressed my comments.

---

## [Author Response]

The following is the authors’ response to the previous reviews.

**Reviewer #1 (Public review):**
Summary:This study builds off prior work that focused on the molecule AA147 and its role as an activator of the ATF6 arm of the unfolded protein response. In prior manuscripts, AA147 was shown to enter the ER, covalently modify a subset of protein disulfide isomerases (PDIs), and improve ER quality control for the disease-associated mutants of AAT and GABAA. Unsuccessful attempts to improve the potency of AA147 have led the authors to characterize a second hit from the screen in this study: the phenylhydrazone compound AA263. The focus of this study on enhancing the biological activity of the AA147 molecule is compelling, and overcomes a hurdle of the prior AA147 drug that proved difficult to modify. The study successfully identifies PDIs as a shared cellular target of AA263 and its analogs. The authors infer, based on the similar target hits previously characterized for AA147, that PDI modification accounts for a mechanism of action for AA263.Strengths:The authors are able to establish that, like AA147, AA263 covalently targets ER PDIs. The work establishes the ability to modify the AA263 molecule to create analogs with more potency and efficacy for ATF6 activation. The "next generation" analogs are able to enhance the levels of functional AAT and GABAA receptors in cellular models expressing the Z-variant of AAT or an epilepsy-associated variant of the GABAA receptor, outlining the therapeutic potential for this molecule and laying the foundation for future organism-based studies.

We thank the reviewer for the positive comments on our manuscript. We address the reviewers remaining comments on our work, as described below.

Weaknesses:Arguably, the work does not fully support the statement provided in the abstract that the study "reveals a molecular mechanism for the activation of ATF6". The identification of targets of AA263 and its analogs is clear. However, it is a presumption that the overlap in PDIs as targets of both AA263 and AA147 means that AA263 works through the PDIs. While a likely mechanism, this conclusion would be bolstered by establishing that knockdown of the PDIs lessens drug impact with respect to ATF6 activation.

We thank the reviewer for this comment. We previously showed that genetic depletion of different PDIs modestly impacts ATF6 activation afforded by ATF6 activating compound such as AA147 (see Paxman et al (2018) ELIFE). However, as discussed in this manuscript, the ability for AA147 and AA263 to activate ATF6 signaling is mediated through polypharmacologic targeting of multiple different PDIs involved in regulating the redox state of ATF6. Thus, individual knockdowns are predicted to only minimally impact the ability for AA263 and its analogs to activate ATF6 signaling.

To address this comment, we have tempered our language regarding the mechanism of AA263-dependent ATF6 activation through PDI targeting described herein to better reflect the fact that we have not explicitly proven that PDI targeting is responsible for this activity, as highlighted below:

“Page 7, Line 158: “Intriguingly, 12 proteins were shared between these two conditions, including 7 different ER-localized PDIs (Fig. 1H). This includes PDIs previously shown to regulate ATF6 activation including TXNDC12/ERP18.[45,46] These results are similar to those observed when comparing proteins modified by the selective ATF6 activating compound AA147^yne^ and AA132^yne^.[38] Further, we found that the extent of labeling for PDIs including PDIA1, PDIA4, PDIA6, and TMX1, but not TXNDC12, showed greater modification by AA132^yne^, as compared to AA263^yne^ (Fig. 1I). Similar results were observed for AA147^yne^.[38] This suggests that, like AA147, the selective activation of ATF6 afforded by AA263 is likely attributed to the modifications of a subset of multiple different ER-localized PDIs by this compound.”

Alternatively, it has previously been suggested that the cell-type dependent activity of AA263 may be traced to the presence of cell-type specific P450s that allow for the metabolic activation of AA263 or cell-type specific PDIs (Plate et al 2016; Paxman et al 2018). If the PDI target profile is distinct in different cell types, and these target difference correlates with ATF6-induced activity by AA263, that would also bolster the authors' conclusion.

As highlighted by the reviewer, different ER oxidases (e.g., P450s) could differentially influence activation of compounds such as AA263 to promote PDI modification and subsequent ATF6 activation. The specific ER oxidases responsible for AA263 activation are currently unknown; however, we anticipate that multiple different enzymes can promote this activity making it difficult to discern the specific contributions of any one oxidase. We have made this point clearer in the revised submission, as below:

Page 7, Line 169: “This specificity for ER proteins instead suggests the localized generation of AA263 quinone methides at the ER membrane, likely through metabolic activation by different ER localized oxidases, which has been previously been shown to contribute to the selective modification of ER proteins afforded by other compounds such as AA147 [49]”

**Reviewer #2 (Public review):**
Modulating the UPR by pharmacological targeting of its sensors (or regulators) provides mostly uncharted opportunities in diseases associated with protein misfolding in the secretory pathway. Spearheaded by the Kelly and Wiseman labs, ATF6 modulators were developed in previous years that act on ER PDIs as regulators of ATF6. However, hurdles in their medicinal chemistry have hampered further development. In this study, the authors provide evidence that the small molecule AA263 also targets and covalently modifies ER PDIs, with the effect of activating ATF6. Importantly, AA263 turned out to be amenable to chemical optimization while maintaining its desired activity. Building on this, the authors show that AA263 derivatives can improve the aggregation, trafficking, and function of two disease-associated mutants of secretory pathway proteins. Together, this study provides compelling evidence for AA263 (and its derivatives) being interesting modulators of ER proteostasis. Mechanistic details of its mode of action will need more attention in future studies that can now build on this.

We thank the reviewer for their positive comments on our manuscript. We address the reviewer’s specific queries on our work, as outlined below.

In detail, the authors provide strong evidence that AA263 covalently binds to ER PDIs, which will inhibit the protein disulfide isomerase activity. ER PDIs regulate ATF6, and thus their finding provides a mechanistic interpretation of AA263 activating the UPR. It should be noted, however, that AA263 shows broad protein labeling (Figure 1G), which may suggest additional targets, beyond the ones defined as MS hits in this study.

This is true. We do show broad proteome-wide labeling with AA263^yne^, which are largely reflected in the hits identified by MS beyond PDI family members. It is possible that other observed engaged targets, in addition to PDIs, may contribute to the activation of ATF6 signaling. Regardless, our MS analysis clearly shows that the compounds modified by AA263 are enriched for PDIs, further supporting our model whereby AA263-dependent PDI modification is likely responsible for ATF6 activation.

Also, a further direct analysis of the IRE1 and PERK pathways (activated or not by AA263) would have been a benefit, as e.g., PDIA1, a target of AA263, directly regulates IRE1 (Yu et al., EMBOJ, 2020), and other PDIs also act on PERK and IRE1. The authors interpret modest activation of IRE1/PERK target genes (Figure 2C) as an effect on target gene overlap, indeed the most likely explanation based on their selective analyses on IRE1 (ERdj4) and PERK (CHOP) downstream genes, but direct activation due to the targeting of their PDI regulators is also a possible explanation.

While we do observe mild increases in IRE1/XBP1s target genes, we do not observe significant increases in PERK/ISR target genes in cells treated with optimized AA263 analogs (see Fig. 2C). We previously showed that genetic ATF6 activation leads to a modest increase in IRE1/XBP1s target genes, reflecting the overlap in target genes of the IRE1/XBP1s and ATF6 pathways (see Shoulders et al (2013) Cell Reports). However, with our data, we cannot explicitly rule out the possibility that the mild increase in IRE1/XBP1s target genes reflects direct IRE1/XBP1s activation, as suggested by the reviewer. To address this, we have adapted the text to highlight this point, now specifically referring to preferential ATF6 activation afforded by these compounds, as below:

Page 5, Line 100: “In addition to finding AA147, our original high-throughput screen also identified the phenylhydrazone compound AA263 as a compound that preferentially activates the ATF6 arm of the UPR [26]”

Further key findings of this paper are the observed improvement of AAT behavior and GABAA trafficking and function. Further strength to the mechanistic conclusion that ATF6 activation causes this could be obtained by using ATF6 inhibitors/knockouts in the presence of AA263 (as the target PDIs may directly modulate the behavior of AAT and/or GABAA).

AA263 and related compounds could influence ER proteostasis of destabilized proteins through multiple mechanisms including ATF6 activation or direct modification of a subset of PDIs. We previously showed that AA263-dependent enhancement of A1AT-Z secretion and activity can be largely attributed to ATF6 activation (see Sun et al (2023) Cell Chem Biol). In the revised submission, we now show that increased levels of g2(R177G) afforded by treatment with AA263^yne^ are partially blocked by co-treatment with the ATF6 inhibitor Ceapin-A7 (CP7), highlighting the contributions of ATF6 activation for this phenotype (Fig. S5B,C). Intriguingly, this result also demonstrates the benefit for targeting ER proteostasis using compounds such as our optimized AA263 analogs, as this approach allows us to enhance ER proteostasis of destabilized proteins through multiple mechanisms. We further expand on this specific point in the revised manuscript as below:

Page 14, Line 375: “AA263 and its related analogs can influence ER proteostasis in these models through different mechanisms including ATF6-dependent remodeling of ER proteostasis and direct alterations to the activity of specific PDIs.(*) Consistent with this, we show that pharmacologic inhibition of ATF6 only partially blocks increases of g2(R177G) afforded by treatment with AA263^yne^, highlighting the benefit for targeting multiple aspects of ER proteostasis to enhance ER proteostasis of this diseaserelevant GABA_A_ variant. While additional studies are required to further deconvolute the relative contributions of these two mechanisms on the protection afforded by our optimized compounds, our results demonstrate the potential for these compounds to enhance ER proteostasis in the context of different protein misfolding diseases.”

Along the same line, it also warrants further investigation why the different compounds, even if all were used at concentrations above their EC50, had different rescuing capacities on the clients.

This is an interesting question that we are continuing to study. While in general, we observe fairly good correlation between ATF6 activation and correction of diseases of ER proteostasis linked to proteins such as A1AT-Z or GABA_A_ receptors, as the reviewer points out, we do find some compounds are more efficient at correcting proteostasis than others activate ATF6 to similar levels. We attribute this to differences in either labeling efficiency of PDIs or differential regulation of various ER proteostasis factors, although that remains to be further defined. As we continue working with these (and other) compounds, we will focus on defining a more molecular basis for these findings.

Together, the study now provides a strong basis for such in-depth mechanistic analyses.

We agree and we are continuing to pursue the mechanistic basis of ER proteostasis remodeling afforded by these and related compounds.

**Reviewer #3 (Public review):**
Summary:This study aims to develop and characterize phenylhydrazone-based small molecules that selectively activate the ATF6 arm of the unfolded protein response by covalently modifying a subset of ER-resident PDIs. The authors identify AA263 as a lead scaffold and optimize its structure to generate analogs with improved potency and ATF6 selectivity, notably AA263-20. These compounds are shown to restore proteostasis and functional expression of disease-associated misfolded proteins in cellular models involving both secretory (AAT-Z) and membrane (GABAA receptor) proteins. The findings provide valuable chemical tools for modulating ER proteostasis and may serve as promising leads for therapeutic development targeting protein misfolding diseases.Strengths:(1) The study presents a well-defined chemical biology framework integrating proteomics, transcriptomics, and disease-relevant functional assays.(2) Identification and optimization of a new electrophilic scaffold (AA263) that selectively activates ATF6 represents a valuable advance in UPR-targeted pharmacology.(3) SAR studies are comprehensive and logically drive the development of more potent and selective analogs such as AA263-20.(4) Functional rescue is demonstrated in two mechanistically distinct disease models of protein misfolding-one involving a secretory protein and the other a membrane protein-underscoring the translational relevance of the approach.

We thank the reviewer for their positive comments related to our work. We address specific weaknesses highlighted by the reviewer, as outlined below.

Weaknesses:(1) ATF6 activation is primarily inferred from reporter assays and transcriptional profiling; however, direct evidence of ATF6 cleavage is lacking.

While ATF6 trafficking and processing can be visualized in cell culture models following severe ER insults (e.g., Tg, Tm), we showed previously that the more modest activation afforded by pharmacologic activators such as AA147 and AA263 cannot be easily visualized by monitoring ATF6 processing (see Plate et al (2016) ELIFE). As we have shown in numerous other manuscripts, we have established a transcriptional profiling approach that accurately defines ATF6 activation. We use that approach to confirm preferential ATF6 activation in this manuscript. We feel that this is sufficient for confirming ATF6 activation. However, we also now include data showing that co-treatment with ATF6 inhibitors (e.g., CP7) blocks increased expression of ATF6 target genes induced by our prioritized compound AA263^yne^ (Fig. S1B). This further supports our assertion that this compound activates ATF6 signaling.

(2) While the mechanism involving PDI modification and ATF6 activation is plausible, it remains incompletely characterized.

We thank the reviewer for this comment. We previously showed that genetic depletion of different PDIs modestly impacts ATF6 activation afforded by ATF6 activating compound such as AA147. However, as discussed in this manuscript, the ability for AA147 and AA263 to activate ATF6 signaling is mediated through polypharmacologic targeting of multiple different PDIs involved in regulating ATF6 redox. Thus, individual knockdowns are predicted to only minimally impact the ability for AA263 and its analogs to activate ATF6 signaling.

To address this comment, we have tempered out language regarding the mechanism of AA263-dependent ATF6 activation through PDI targeting described herein to better reflect the fact that we have not explicitly proven that PDI targeting is responsible for this activity, as highlighted below:

Page 7, Line 158: “Intriguingly, 12 proteins were shared between these two conditions, including 7 different ER-localized PDIs (Fig. 1H). This includes PDIs previously shown to regulate ATF6 activation including TXNDC12/ERP18.[45,46] These results are similar to those observed when comparing proteins modified by the selective ATF6 activating compound AA147^yne^ and AA132^yne^.[38] Further, we found that the extent of labeling for PDIs including PDIA1, PDIA4, PDIA6, and TMX1, but not TXNDC12, showed greater modification by AA132^yne^, as compared to AA263^yne^ (Fig. 1I). Similar results were observed for AA147^yne^[38] This suggests that, like AA147, the selective activation of ATF6 afforded by AA263 is likely attributed to the modifications of a subset of multiple different ER-localized PDIs by this compound.”

(3) No in vivo data are provided, leaving the pharmacological feasibility and bioavailability of these compounds in physiological systems unaddressed.

We are continuing to test the in vivo activity of these compounds in work outside the scope of this initial study.

**Reviewer #1 (Recommendations for the authors):**
(1) First page of the discussion, last sentence. "We previously showed the relatively labeling of PDI modification directly impacts..." should be reworded.

Thank you. We have corrected this in the revised manuscript.

(2) What is the rationale for measuring ERSE-Fluc activity at 18 h but RNAseq at 6 h? What is known about the timing of action for AA263?

Compound-dependent activation of luciferase reporters requires the translation and accumulation of the luciferase protein for sufficient signal, while qPCR does not. We normally use longer incubations for reporter assays to ensure that we have sufficient quantity of reporter protein to accurately monitor activation. We have found that AA263 can rapidly increase ATF6 activity, with gene expression increases being observed after only a few hours of treatment. This is consistent with the proposed mechanism of ATF6 activation discussed herein involving metabolic activation and subsequent PDI modification.

(3) Figure 1 panel E and Figure S2 panel B. Are these the same data for AA263 and AA263yne, with the AA2635 added to the plot for Figure S2? If so, it would be nice to note that panel B represents data from 3 of the replicates that are shown in Figure 1 (n=6).

Yes. The AA263 and AA263^yne^ data shown in Fig. 1E and Fig. S2B are the same data, as these experiments were performed at the same time. We apologize for this oversight, which has now been corrected in the revised version. Note that there were n=3 replicates for the dose response shown in Fig. 1E, which we corrected in the figure legend as below:

Fig. S2B Figure Legend: “B. Activation of the ERSE-FLuc ATF6 reporter in HEK293T cells treated for 18 h with the indicated concentration of AA263, AA263^yne^, or AA263-5. Error bars show SEM for n = 3 replicates. The data for AA263 and AA263^yne^ is the same as that shown in Fig. 1E and are shown for comparison.”

(4) Figure S3. The legend notes 5 µM AA263-yne and 20 µM analog, whereas the figure itself outlines the same ratio but different concentrations: 10 µM and 40 µM.

We apologize for this mistake in the legend, which has been corrected. The information in the figure is correct.

**Reviewer #2 (Recommendations for the authors):**
(1) The activation mechanism of ATF6 is still debated (really trafficking as a monomer?); the authors may want to word more carefully here.

We agree. We have corrected this in the revised manuscript to indicate that increased populations of reduced ATF6 traffic for proteolytic processing.

(2) In Figure 1B, below the figure, mM is written for BME, but micromolar is meant.

Thank you. This has been corrected in the revised manuscript.

(3) The authors may want to make clearer, why BME does not completely inhibit AA263 and does not cause ER stress itself under the conditions tested.

The addition of BME in our experiments is designed to shift the redox potential of the cell to increase intracellular thiol reagents, such as glutathione, that can quench ‘activated’ AA263 and its analogs. However, BME is actively being oxidized upon addition and the intracellular redox environment can rapidly equilibrate following BME addition. Thus, we do not expect that AA263 or other metabolically activated compounds will be fully quenched using this approach, as is observed. This is consistent with other experiments where we show that the use of these types of reducing agents do not fully suppress the activity of reactive molecules, instead shifting their dosedependent activation of specific pathways.

(4) The data in Figure 4C seems to disagree with the other data on the tested compounds; this should be clarified.

It is unclear to what the reviewer is referring. The data in 4C shows that treatment with our optimized AA263 analogs improved elastase inhibition afforded by secreted A1AT, as would be predicted.

(5) PDIs that have been shown to regulate ATF6 should be discussed in more detail in the light of the presented data/interactome (e.g., ERp18).

Thank you for the suggestion. We now explicitly note that AA263^yne^ covalent modifies TXNDC12/ERP18 in our proteomic dataset. However, we also note that there is no difference in labeling of this specific PDI between AA263^yne^ and AA132^yne^. This may indicate that the targeting of this protein is responsible for the larger levels of ATF6 activation afforded by both these compounds relative to AA147, with the activation of other UPR pathways afforded by AA132 resulting from increased labeling of other PDIs. We are now exploring this possibility in work outside the scope of this current manuscript.

Page 7 Line 158: “Intriguingly, 12 proteins were shared between these two conditions, including 7 different ER-localized PDIs (Fig. 1H). This includes PDIs previously shown to regulate ATF6 activation including TXNDC12/ERP18.[45,46] These results are similar to those observed when comparing proteins modified by the selective ATF6 activating compound AA147^yne^ and AA132^yne^.[38] Further, we found that the extent of labeling for PDIs including PDIA1, PDIA4, PDIA6, and TMX1, but not TXNDC12, showed greater modification by AA132^yne^, as compared to AA263^yne^ (Fig. 1I). Similar results were observed for AA147^yne^ [38] This suggests that, like AA147, the selective activation of ATF6 afforded by AA263 is likely attributed to the modifications of a subset of multiple different ER-localized PDIs by this compound.”

**Reviewer #3 (Recommendations for the authors):**
(1) Please consider adding detection of ATF6 cleavage by Western blot as direct evidence of AA263-induced ATF6 activation, to substantiate the central mechanistic claim.

While ATF6 trafficking and processing can be visualized in cell culture models following severe ER insults (e.g., Tg, Tm), we showed previously that the more modest activation afforded by pharmacologic activators such as AA147 and AA263 cannot be easily visualized through monitoring ATF6 proteolytic processing by western blotting (see Plate et al (2016) ELIFE). As we have shown in numerous other manuscripts, we have established a transcriptional profiling approach that accurately defines ATF6 activation. We use that approach to confirm preferential ATF6 activation in this manuscript. We feel that this is sufficient for confirming ATF6 activation. However, we also now include qPCR data showing that co-treatment with ATF6 inhibitors (e.g., CP7) blocks increased expression of ATF6 target genes induced by our prioritized compounds.

(2) To strengthen causal inference, loss-of-function experiments such as PDI knockdown, cysteine mutant inactivation, or reconstitution studies may be informative.

We thank the reviewer for this comment. We previously showed that genetic depletion of different PDIs modestly impacts ATF6 activation afforded by ATF6 activating compound such as AA147. However, as discussed in this manuscript, the ability for AA147 and AA263 to activate ATF6 signaling is mediated through polypharmacologic targeting of multiple different PDIs involved in regulating ATF6 redox state rather than a single PDI family member. Thus, individual knockdowns are predicted to only minimally impact the ability for AA263 and its analogs to activate ATF6 signaling.

To address this comment, we have tempered out language regarding the mechanism of AA263-dependent ATF6 activation through PDI targeting described herein to better reflect the fact that we have not explicitly proven that PDI targeting is responsible for this activity.

(3) Since β-mercaptoethanol inhibits ATF6 activation, it would be helpful to examine whether DTT also suppresses the activity of AA263 or its analogs, to clarify the redox sensitivity of the mechanism.

The use of reducing agents stronger than BME, such as DTT, globally activates the UPR, including the ATF6 arm of the UPR. Thus, we are unable to perform the requested experiments. We specifically use BME because it is a sufficiently mild reducing agent that can quench reactive metabolites (e.g., activated AA263 analogs) through alterations in cellular glutathione levels without globally activating the UPR.

(4) Given the electrophilic nature of AA263, which may allow it to react with endogenous thiols (e.g., glutathione or cysteine), a brief discussion or experimental validation of this potential liability would enhance the interpretation of in vivo applicability.

Metabolically activated AA263, like AA147, can be quenched by endogenous thiols such as glutathione. However, treatment with our metabolically activatable electrophiles AA147 and AA263 , either in vitro or in vivo, does not seem to induce activation of the NRF2-regulated oxidative stress response (OSR) in the cell lines used in this manuscript (e.g., Fig. S2C). This suggests that treatment with these compounds does not globally disrupt the intracellular redox state, at least in the tested cell lines. While AA147 has been shown to activate NRF2 in specifical neuronal cell lines and in primary neurons, AA147 does not activate NRF2 signaling in other nonneuronal cell lines or other tissues (see Rosarda et al (2021) ACS Chem Bio). We are currently testing the potential for AA263 to similarly activate adaptive NRF2 signaling in neuronal cells. Regardless, AA147, which functions through a similar mechanism to that proposed for AA263, has been shown to be beneficial in multiple models of disease both in vitro and in vivo. This indicates that this mechanism of action is suitable for continued translational development to mitigate pathologic ER proteostasis disruption observed in diverse types of human disease.

(5) Evaluation of in vivo activity, such as BiP induction in the liver following intraperitoneal administration of AA263-20 or related analogs, could substantially increase the translational impact of the work.

We are continuing to probe the activity of our optimized AA263 analogs in vivo in work outside the scope of this current manuscript. We thank the reviewer for this suggestion.

(6) The degree of BiP induction may also be contextualized by comparison with known ER stress inducers such as thapsigargin or tunicamycin, ideally by providing relative dose-equivalent responses.

We are not sure to what the reviewer is referring. We show comparative activation of ATF6 in cells treated with the ER stressor Tg and our compounds by both reporter assay (e.g., Fig. 2B) and qPCR of the ATF6 target gene BiP (HSPA5) (Fig. S2A). We feel that this provides context for the more physiologic levels of ATF6 activation afforded by these compounds.